# Structured Prompting Enables More Robust Evaluation of Language Models

## Abstract

As language models (LMs) are increasingly adopted across domains, high-quality benchmarking frameworks that accurately estimate performance are essential for guiding deployment decisions. While frameworks such as Holistic Evaluation of Language Models (HELM) enable broad evaluation across tasks, they often rely on fixed prompts that fail to generalize across LMs, yielding unrepresentative performance estimates. Unless we approximate each LM's ceiling (maximum achievable via changes to the prompt), we risk underestimating performance. Declarative prompting frameworks, such as DSPy, offer a scalable alternative to manual prompt engineering by crafting structured prompts that can be optimized per task. However, such frameworks have not been systematically evaluated across established benchmarks. We present a reproducible *DSPy+HELM* framework that introduces structured prompting methods which elicit reasoning, enabling more accurate LM benchmarking. Using four prompting methods, we evaluate four frontier LMs across seven benchmarks (general/medical domain) against existing HELM baseline scores. We find that without structured prompting: (i) HELM underestimates LM performance (by 4% average), (ii) performance estimates vary more across benchmarks (+2% standard deviation), (iii) performance gaps are misrepresented (leaderboard rankings flip on 3/7 benchmarks), and (iv) introducing reasoning (*chain-of-thought*) reduces LM sensitivity to prompt design (smaller performance $\Delta$ across prompting methods). To our knowledge, this is the first benchmarking study to systematically integrate structured prompting into an established evaluation framework, demonstrating how scalable performance-ceiling approximation yields more robust, decision-useful benchmarks. We open-source (i) *DSPy+HELM* Integration[1] and (ii) Prompt Optimization Pipeline[2].

## 1 Introduction

Language models (LMs) have rapidly advanced in text generation, spurring deployment across diverse domains (Thirunavukarasu et al., 2023; Van Veen et al., 2024; Seo et al., 2024). Yet, integrating LMs into downstream workflows remains challenging as LMs frequently commit errors (Aali et al., 2025). Even state-of-the-art general-purpose frontier LMs exhibit non-trivial hallucination rates (Wang et al., 2024a; Sivarajkumar et al., 2024; Bang et al., 2025; Tamber et al., 2025). Such concerns are compounded by LMs' sensitivity to prompt design (Razavi et al., 2025), introducing variability in leaderboard performance.

While benchmarking frameworks such as Holistic Evaluation of Language Models (HELM) (Liang et al., 2022; Bedi et al., 2025) enable holistic evaluation via a comprehensive suite covering diverse tasks, public leaderboards typically evaluate multiple LMs under a fixed prompt per benchmark. However, fixed prompts rarely generalize well across LMs, leading to unrepresentative performance estimates that obscure underlying strengths and weaknesses of LMs. Hence, broader LM adoption necessitates scalable approximation of performance ceilings (i.e., the maximum achievable via prompt-only changes), thereby allowing practitioners to weigh cost–benefit tradeoffs and choose the right model for each downstream task.

---

[1]*DSPy+HELM* Integration: `https://anonymous.4open.science/pr/8684`
[2]Prompt Optimization Pipeline: `https://anonymous.4open.science/r/dspy-helm`

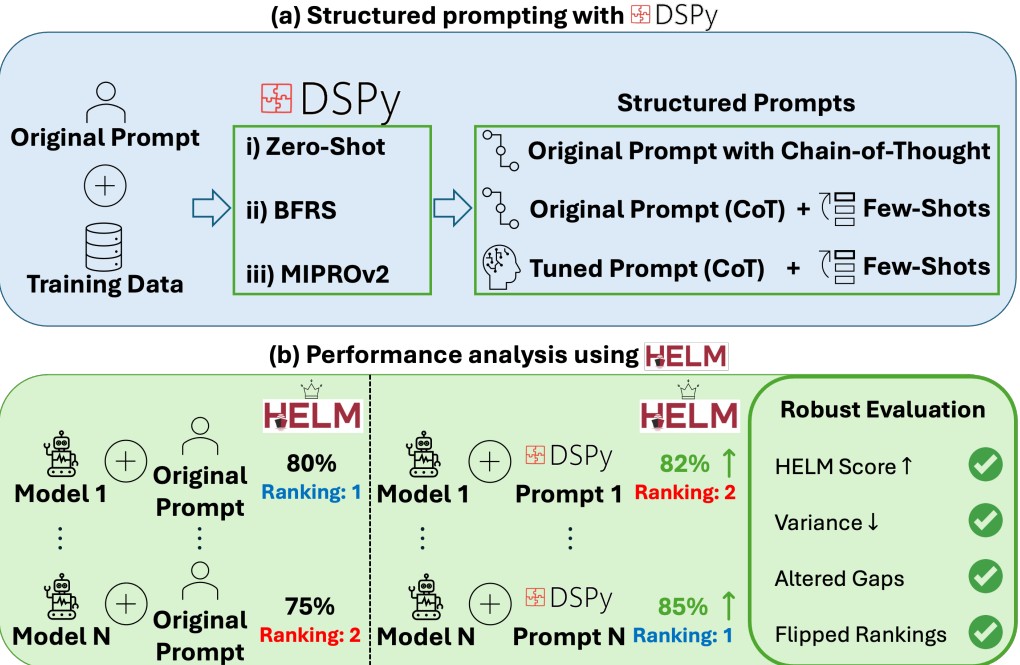

Figure 1: Pipeline overview. (a) DSPy takes HELM's baseline prompt and produces structured prompt variants. (b) HELM evaluates models under each prompt variant. With structured prompting, we observe more robust evaluation: (i) improved performance, (ii) reduced variance, (iii) altered gaps (flipped rankings).

Prompt engineering has emerged as a practical alternative to fine-tuning. Well-designed prompts improve performance, as demonstrated by Nori et al. (2024); Maharjan et al. (2024), combining few-shot selection, chain-of-thought (CoT) (Wei et al., 2022), and ensembling. However, these methods rely on hand-engineered prompts, demanding domain expertise and iterative experimentation, making them labor-intensive and often non-robust to new model rollouts (Wang et al., 2025). Consequently, researchers have explored automatic prompt optimization (APO) (Li et al., 2025), which treats prompt design as an optimization problem.

DSPy (Khattab et al., 2023) is a widely used declarative framework that represents prompts as modular, parameterized components with an intuitive structure that allows moving from zero-shot prompts to more adaptive prompting styles all within a single unified system supporting reproducible, structured prompting. Moreover, DSPy supports automatic prompt optimizers (APOs) such as MIPROv2 (Opsahl-Ong et al., 2024), which can convert high-level task specifications into optimized instructions and few-shot examples.

However, despite the growing use of structured prompting, we lack a systematic evaluation of how these approaches affect benchmark robustness and performance estimates across established evaluation suites. We use DSPy as an instantiation of structured prompting and integrate it with HELM (Figure 1), presenting:

1. A reproducible *DSPy+HELM* framework that introduces structured prompting methods which elicit reasoning, enabling more robust evaluation of LMs across HELM benchmarks.

2. An evaluation of prompting methods (Zero-Shot, Bootstrap Few-Shot with Random Search, MIPROv2) against HELM's baseline across four LMs and seven HELM benchmarks that span general and medical domains (reasoning, knowledge QA, problem-solving, error-classification), where each prompting method leverages a distinct mechanism for refining prompts to approximate LM performance ceilings.

3. Empirical evidence that without structured prompting: (i) HELM underestimates LM performance (by 4% average), (ii) performance estimates vary more across benchmarks (+2% standard deviation), (iii) performance gaps are misrepresented (leaderboard rankings flip on 3/7 benchmarks), and (iv) introducing reasoning (CoT) reduces LM sensitivity to prompt design (smaller $\Delta$ across prompts).

**Prompt 1: HELM Baseline**

*Given a patient note and a clinical question, compute the requested medical value.*
*Patient Note and Question: ———————————————————————*

**Prompt 2: Zero-Shot CoT**

Your input fields are: "INPUTS"
Your output fields are: "REASONING" and "OUTPUT"
Your objective is: Given the fields "INPUTS", produce the fields "OUTPUT"

INPUTS:

*Given a patient note and a clinical question, compute the requested medical value.*
*Patient Note and Question: ———————————————————————*

Respond with the corresponding output fields, starting with "REASONING", then "OUTPUT".

**Prompt 3: BFRS (Few-Shot Optimized)**

Your input fields are: "INPUTS"
Your output fields are: "REASONING" and "OUTPUT"
Your objective is: Given the fields "INPUTS", produce the fields "OUTPUT"

IN-CONTEXT EXAMPLES ($K$ Demos):

INPUTS: <input text> → REASONING: <steps>, OUTPUT: <output text>

INPUTS:

*Given a patient note and a clinical question, compute the requested medical value.*
*Patient Note and Question: ———————————————————————*

Respond with the corresponding output fields, starting with "REASONING", then "OUTPUT".

**Prompt 4: MIPROv2 (Instruction + Few-Shot Optimized)**

Your input fields are: "INPUTS"
Your output fields are: "REASONING" and "OUTPUT"
Your objective is: You are a highly skilled medical expert working in a busy emergency room. A patient presents with a complex medical history and concerning symptoms. The attending physician needs your immediate assistance in calculating a critical risk score to guide treatment decisions. The patient's life may depend on your accuracy.

IN-CONTEXT EXAMPLES ($K$ Demos):

INPUTS: <input text> → REASONING: <steps>, OUTPUT: <output text>

INPUTS:

*Given a patient note and a clinical question, compute the requested medical value.*
*Patient Note and Question: ———————————————————————*

Respond with the corresponding output fields, starting with "REASONING", then "OUTPUT".

Figure 2: Structured prompting methods evaluated in our study (Zero-Shot CoT, BFRS, MIPROv2). Each box corresponds to one method, showing how instructions and context differ across methods. For BFRS and MIPROv2, $K$ denotes the number of in-context demonstrations (Inputs → Reasoning, Output).

| Model | API Identifier | Release | Context | Reasoning |
|---|---|---|---|---|
| Claude 3.7 Sonnet | `anthropic/claude-3-7-sonnet-20250219` | 02/19/2025 | 200k | ✗ |
| Gemini 2.0 Flash | `google/gemini-2.0-flash-001` | 02/01/2025 | 1000k | ✗ |
| GPT 4o | `openai/gpt-4o-2024-05-13` | 05/13/2024 | 128k | ✗ |
| o3 Mini | `openai/o3-mini-2025-01-31` | 01/31/2025 | 200k | ✓ |

Table 1: Language models evaluated in our study. Columns show API identifiers, release dates, maximum context windows, and native reasoning modes (yes/no). We choose widely used models to evaluate whether prompting meaningfully affects even top-tier frontier models. All experiments were run in August 2025.

## 2 Methodology

DSPy (Khattab et al., 2023) is a framework for composing modular LM pipelines. Formally, let $\Phi$ denote a LM program with $m$ modules. Each module $i$ has a prompt template $p_i$ containing a set of variables (open slots) for the instruction and $K$ demonstration examples. Let $V$ be the set of all such prompt variables across $\Phi$, and let $V \to S$ denote an assignment of each variable $v \in V$ to a concrete string $s \in S$. We write $\Phi_{V \to S}$ to denote running program $\Phi$ under a particular prompt assignment. Given a dataset $D = (x, y)$ of inputs $x$ with ground-truth $y$ and an evaluation metric $\mu$ that compares the program's output $\Phi(x)$ against $y$, the optimization maximizes $\mu$ over all instructions and demonstrations:

$$\Phi^* = \underset{V \to S}{\arg\max} \frac{1}{|D|} \sum_{(x,y) \in D} \mu\Big(\Phi_{V \to S}(x), y\Big). \tag{1}$$

### 2.1 Prompting Methods

**Baseline Prompting**

As a baseline, we evaluate LMs using the following prompting methods:

**1. HELM Baseline.** HELM supports multiple prompting configurations; we adopt the commonly reported fixed, zero-shot (hand-crafted) prompt configuration without CoT as the baseline for comparison.

**2. Zero-Shot Predict.** DSPy's Zero-Shot Predict configuration is an unoptimized non-adaptive baseline, which we instantiate with the `dspy.Predict` module. Each module's instruction prompt is initialized with the same HELM baseline instruction, without in-context demonstrations (i.e. $K = 0$).

**Structured Prompting**

In addition, we evaluate LMs using the following structured prompting methods (Figure 2):

**1. Zero-Shot CoT.** DSPy's Zero-Shot CoT configuration utilizes the same prompting structure as Zero-Shot Predict, but instead instantiates the `dspy.ChainOfThought` module, which elicits step-by-step rationales, instructing the LM to generate an explicit reasoning trace with the output.

**2. BFRS.** Bootstrap Few-Shot with Random Search (BFRS) (Algorithm 1) leverages the idea of bootstrapping and random sampling to select the best few-shot demonstrations (fixed instructions) in two phases: (i) Bootstrapping demonstrations: the LM program $\Phi$ is run on a subset of training inputs to gather traces for each module. Whenever the output of $\Phi(x)$ for an example $x$ achieves a sufficiently high score (on metric $\mu$), the input-output pair is taken as a candidate demonstration. (ii) Random few-shot search: Given demonstration pools, BFRS randomly samples sets of $K$ demonstrations per module, inserts them into the module, and evaluates the program on a validation split. After trying $N$ combinations, the program with the highest score is returned (with hyperparameters $K$ and $N$).

| Benchmark | Input → Output | Task | Samples |
|---|---|---|---|
| MMLU-Pro | Reasoning Question → Answer | Multi-Task Reasoning | 1,000 |
| GPQA | Graduate Question → Answer | Graduate-Level QA | 446 |
| GSM8K | Math Problem → Solution | Numeric Problem-Solving | 1,000 |
| MedCalc-Bench | Patient Note → Computed Value | Computational Reasoning | 1,000 |
| Medec | Medical Narrative → Errors | Error Classification | 597 |
| HeadQA | Medical Question → Answer | USMLE-Style QA | 1,000 |
| MedBullets | Medical Question → Answer | USMLE-Style QA | 308 |

Table 2: HELM benchmarks (publicly available) evaluated in our study. Columns summarize each benchmark's input → output mapping, underlying task type, and number of test samples. The benchmarks span reasoning, knowledge QA, problem-solving, and error-classification tasks across both general and medical domains.

**3. MIPROv2.** MIPROv2 (Algorithm 2) is an optimizer that jointly selects instructions and $K$ few-shot demonstrations via: (i) bootstrapping demos, (ii) grounded instruction proposals from a proposer LM conditioned on dataset summaries, program structure, exemplar demos, and trial history, and (iii) Bayesian search over instruction-demo pairs. It treats each configuration $\mathbf{v}$ as hyperparameters, learns $p(y \mid \mathbf{v})$ from trial outcomes, and steers toward high-scoring regions. For efficiency, candidates are scored on mini-batches of size $B$, with periodic full-dataset $D$ evaluations of top contenders; the best full-data configuration is returned (with hyperparameters instruction text, demo-set, and $K$).

## 2.2 Benchmarks

We choose seven benchmarks (Table 2) based on (i) public availability, (ii) task diversity (reasoning, knowledge QA, problem-solving, error classification), and (iii) domain coverage (general/medical).

**MMLU-Pro.** MMLU-Pro (Wang et al., 2024b) is an enhanced version of MMLU that focuses on more challenging, reasoning-intensive questions. It expands answer choices from four to ten and removes trivial items, providing a more discriminative measure of higher-order reasoning. The metric $\mu$ is exact match.

**GPQA.** GPQA (Rein et al., 2024) is a graduate-level multiple-choice benchmark covering biology, physics, and chemistry to test advanced reasoning. The metric $\mu$ is the fraction of correct answers (exact match).

**GSM8K.** GSM8K (Cobbe et al., 2021) consists of grade school math word problems designed to evaluate reasoning. The task requires computing a final numeric answer, and the metric $\mu$ is exact match.

**MedCalc-Bench.** MedCalc-Bench (Khandekar et al., 2024) is a medical calculation benchmark, where the input is a patient note and a question asking for a numerical/categorical value. The evaluation metric $\mu$ is exact match for the *risk*, *severity*, and *diagnosis* categories, and a within-range correctness check for others.

**Medec.** Medec (Abacha et al., 2024) is an error detection and correction benchmark, where each input contains a narrative that may contain factual errors, and the task is to identify/correct these errors. The evaluation metric $\mu$ involves checking how accurately LMs identify whether a note contains an error (binary).

**HeadQA.** HeadQA (Vilares & Gómez-Rodríguez, 2019) is a collection of biomedical multiple-choice questions for testing medical knowledge, where questions cover medical knowledge and often resemble medical board exams. The performance metric $\mu$ is exact match between the prediction and the correct option.

**MedBullets.** MedBullets (Medbullets, 2025) is a benchmark of USMLE-style medical questions with multiple-choice answers. MedBullets covers broad topics and is designed to reflect the difficulty of medical licensing exams. Like HeadQA, the primary metric $\mu$ is exact match accuracy on the correct answer.

---

**Algorithm 1** BFRS: Bootstrap Few-Shot with Random Search

---

**Require:** Seed program $\Phi_{\text{seed}}$; train/val sets $D_{\text{tr}}, D_{\text{val}}$; threshold $\tau$; demos per module $K_i$; trials $R$; minibatch size $B$.

1: **Bootstrap:** For each $(x, y) \in D_{\text{tr}}$: run $\Phi_{\text{seed}}$; if $\mu(\Phi_{\text{seed}}(x), y) \geq \tau$, then for each module $i$ add $\left(u_i(x), \Phi_{\text{seed}}^{(i)}(u_i(x))\right)$ to $\mathcal{B}_i$.

2: **Search:** For $r = 1{:}R$:

3: **for** $i = 1{:}m$ **do**

4:     Sample $S_i^{(r)} \leftarrow \text{SampleK}(\mathcal{B}_i, K_i)$

5:     Let $\mathbf{v}^{(r)} \leftarrow (I_1^{\text{seed}}, S_1^{(r)}, \ldots, I_m^{\text{seed}}, S_m^{(r)})$.

6:     Draw minibatch $\mathcal{B} \subset D_{\text{val}}$ with $|\mathcal{B}| = B$; compute $\widehat{J}_B(\mathbf{v}^{(r)})$ by equation 15.

7: **Select:** $\mathbf{v}^\star \in \arg\max_r \widehat{J}_B(\mathbf{v}^{(r)})$; optionally re-evaluate $J(\mathbf{v}^\star)$ on full $D_{\text{val}}$.

8: **Return** $\mathbf{v}^\star$ and the resulting $\Phi_{\mathbf{v}^\star}$.

---

**Algorithm 2** MIPROv2: Joint Optimization of Instructions & Demos

---

**Require:** Train/val sets $D_{\text{tr}}, D_{\text{val}}$; candidate sizes $T_i$ (instructions), $K_i$ (demos per module); minibatch size $B$; escalation period $E$; TPE quantile $\gamma$; trials $T$.

1: **Bootstrap demos:** Build $\{\mathcal{B}_i\}_{i=1}^m$ as in equation 13.

2: **Propose instructions:** For each $i$, sample $\mathcal{I}_i = \{I_i^{(t)}\}_{t=1}^{T_i}$ from proposer LM using task/program-aware context.

3: Initialize history $\mathcal{H}_0 \leftarrow \varnothing$; best full-eval $(\mathbf{v}^\dagger, J^\dagger) \leftarrow (\text{seed}, 0)$.

4: **for** $t = 1{:}T$ **do**

5:     Fit/update TPE from $\mathcal{H}_{t-1}$ to obtain $\ell, g$ in equation 16.

6:     **Acquire candidate:**

$$\mathbf{v}^{(t)} \in \arg\max_{\mathbf{v} \in \prod_i (\mathcal{I}_i \times \mathcal{B}_i^{K_i})} \frac{\ell(\mathbf{v})}{g(\mathbf{v})}.$$

7:     Draw minibatch $\mathcal{B} \subset D_{\text{val}}$, $|\mathcal{B}| = B$, score $y^{(t)} = \widehat{J}_B(\mathbf{v}^{(t)})$.

8:     Append to history: $\mathcal{H}_t \leftarrow \mathcal{H}_{t-1} \cup \{(\mathbf{v}^{(t)}, y^{(t)})\}$.

9:     **if** $t \bmod E = 0$ **then**

10:         Select top-$K$ by running mean; evaluate each on full $D_{\text{val}}$ to get $J(\cdot)$.

11:         If any $J(\mathbf{v}) > J^\dagger$ then update $(\mathbf{v}^\dagger, J^\dagger) \leftarrow (\mathbf{v}, J(\mathbf{v}))$.

12: **Return** $\mathbf{v}^\dagger$ and $\Phi_{\mathbf{v}^\dagger}$.

---

| Prompting Method | Claude 3.7 Sonnet | Gemini 2.0 Flash | GPT 4o | o3 Mini |
|---|---|---|---|---|
| HELM Baseline | 64.81% ± 22.6 | 61.41% ± 23.8 | 61.04% ± 23.9 | 70.93% ± 19.7 |
| Zero-Shot Predict | 65.10% ± 22.6 | 61.69% ± 22.7 | 59.69% ± 25.0 | **73.24% ± 20.3** |
| Zero-Shot CoT | 69.36% ± 18.8 | **66.21% ± 20.9** | 65.67% ± 22.5 | 72.73% ± 19.7 |
| BFRS | 69.34% ± 19.0 | 66.19% ± 21.2 | **65.87% ± 22.9** | 73.07% ± 19.7 |
| MIPROv2 | **69.80% ± 19.0** | 66.19% ± 21.1 | 65.34% ± 23.0 | 73.07% ± 19.6 |
| **Ceiling − Baseline (Δ)** | +4.99% | +4.80% | +4.83% | +2.31% |

Table 3: HELM leaderboard (macro-averaged over seven benchmarks) across four language models and five prompting methods. **Green** marks the "ceiling" performance for a model (best value across prompting methods). Entries are reported as the macro-average ± standard deviation $\sigma$ over seven benchmarks. At each model's ceiling, structured prompting on average leads to +4% in accuracy and −2% in $\sigma$ across benchmarks.

## 2.3 Experimental Setup

**Implementation details.** We evaluate four frontier LMs (Table 1). We initialize each DSPy program with HELM's baseline instruction for comparability. DSPy then applies its own standardized prompting modules, treating the full HELM prompt as input. For BFRS and MIPROv2 optimizers, we follow DSPy's data separation: the demonstration pool is bootstrapped *exclusively* from the training split, while candidate prompts are evaluated on a *disjoint* held-out validation split from the original training partition; neither optimizer ever sees the HELM leaderboard test set. Each benchmark's loader creates a fixed train/val partition (default 90/10 with the same seed), and we cap both bootstrapped and labeled demonstrations at $K \leq 3$ per module. All final scoring is performed via HELM, so outputs are judged identically regardless of how they were produced. All results reflect single, deterministic runs (temperature = 0), matching HELM's experimental setup. For HELM baselines, we report HELM's public leaderboard scores when the setup matches ours: (i) identical LM API version, (ii) zero-shot prompting, and (iii) no CoT reasoning. For benchmarks where the leaderboard setup does not match, we reproduce them with single, deterministic runs.

**Metric calculation.** To summarize gains, we take the mean of the three structured prompting methods (Zero-Shot CoT, BFRS, MIPROv2); for each LM, we first macro-average across benchmarks, and then average the Δ (absolute % change over baseline) across LMs. The change in variability ($\sigma$) is reported analogously.

## 3 Results and Discussion

### 3.1 Impact of Structured Prompting on HELM Leaderboard

**Improved performance over HELM baseline.** Structured prompting methods (Zero-Shot CoT, BFRS, MIPROv2) consistently improve over the HELM baseline (Table 3). On average, LMs gain +4% in absolute accuracy. Non-reasoning models benefit most (+5%), while *o3 Mini* sees smaller but consistent gains (+2%).

**Flipped leaderboard rankings.** At ceiling, three leaderboard rankings flip. On MMLU-Pro (Table 4), baseline *o3 Mini > Claude 3.7 Sonnet* (77.1% vs. 76.3%) reverses to *Claude 3.7 Sonnet > o3 Mini* (80.6% vs. 78.4%). On GSM8K, *GPT 4o* overtakes *Gemini 2.0 Flash*, shifting from (81.1% vs. 84.0%) to (90.7% vs. 84.2%). On MedCalc-Bench (Table 5), baseline *o3 Mini > Claude 3.7 Sonnet* (34.0% vs. 21.0%) becomes *Claude 3.7 Sonnet > o3 Mini* (35.3% vs. 34.7%), highlighting how prompt choice can moderate rankings.

**Altered inter-model performance gaps.** When evaluated at ceiling performance, models can either narrow or widen their relative performance gaps, providing a more accurate view of true capability differences. Averaging across benchmarks, the gap between the top two models (*o3 Mini* and *Claude 3.7 Sonnet*) shrinks from 6% at baseline (70.9 vs. 64.8) to 3% (73.2 vs. 69.8). However, this trend is not uniform: on GPQA, the gap widens substantially, from 0.6% at baseline (57.6 vs. 57.0) to 4.3% at ceiling (68.4 vs. 64.1).

| Benchmark | Prompting Method | Claude 3.7 Sonnet | Gemini 2.0 Flash | GPT 4o | o3 Mini |
|---|---|---|---|---|---|
| MMLU-Pro | HELM Baseline | 76.3% ± 2.7 | 66.1% ± 3.0 | 62.2% ± 3.0 | 77.1% ± 3.1 |
| | Zero-Shot Predict | 77.7% ± 2.6 | 70.3% ± 2.8 | 60.7% ± 3.1 | **78.4% ± 3.1 ↓** |
| | Zero-Shot CoT | 79.7% ± 2.5 | 75.3% ± 2.7 | 67.6% ± 3.0 | 76.2% ± 3.1 |
| | BFRS | 80.1% ± 2.5 | **75.4% ± 2.7** | **71.1% ± 2.8** | 76.5% ± 3.1 |
| | MIPROv2 | **80.6% ± 2.5 ↑** | 75.3% ± 2.7 | 68.7% ± 2.9 | 76.1% ± 3.1 |
| GPQA | HELM Baseline | 57.0% ± 4.7 | 53.4% ± 4.7 | 45.5% ± 4.7 | 57.6% ± 4.5 |
| | Zero-Shot Predict | 62.1% ± 4.5 | 54.5% ± 4.7 | 41.7% ± 4.5 | 66.6% ± 4.3 |
| | Zero-Shot CoT | 61.4% ± 4.7 | 59.2% ± 4.5 | **52.5% ± 4.7** | 66.4% ± 4.5 |
| | BFRS | **64.1% ± 4.5** | **61.0% ± 4.5** | 49.3% ± 4.7 | 65.5% ± 4.5 |
| | MIPROv2 | 61.9% ± 4.5 | 59.0% ± 4.5 | 47.8% ± 4.7 | **68.4% ± 4.3** |
| GSM8K | HELM Baseline | 80.5% ± 2.5 | 84.0% ± 2.3 | 81.1% ± 2.5 | 88.6% ± 2.0 |
| | Zero-Shot Predict | 83.0% ± 2.3 | 77.3% ± 2.6 | 84.6% ± 2.2 | **93.6% ± 1.6** |
| | Zero-Shot CoT | 83.3% ± 2.3 | 83.1% ± 2.4 | **90.7% ± 1.8 ↑** | 92.6% ± 1.7 |
| | BFRS | 83.2% ± 2.3 | **84.2% ± 2.3 ↓** | 90.4% ± 1.9 | 93.0% ± 1.6 |
| | MIPROv2 | **84.0% ± 2.3** | 83.5% ± 2.3 | 89.8% ± 2.0 | 93.4% ± 1.6 |

Table 4: HELM leaderboard (general domain) across four language models and five prompting methods. **Green** marks the "ceiling" performance for a model (best value across prompting methods). ↑ and ↓ indicate a one-step increase or decrease in leaderboard rank, respectively. Entries are reported as mean ± 95% bootstrap confidence interval. Overall, structured prompting consistently improves the robustness of benchmarks.

**Reduced across-benchmark variance.** Structured prompting methods reduce dispersion. Across-benchmark $\sigma$ drops for *Claude 3.7 Sonnet* (22.6% → 18.8%), *Gemini 2.0 Flash* (23.8% → 20.9%), and *GPT 4o* (23.9% → 22.5%), while *o3 Mini* is unchanged (19.7%), indicating lower sensitivity.

**Benchmark-dependent sensitivity.** Performance gains vary across benchmarks (Figure 3). Tasks requiring reasoning, such as MMLU-Pro, GPQA, GSM8K, MedCalc-Bench, and MedBullets, show the largest gains (average +5.5% absolute across models). In contrast, HeadQA and Medec exhibit smaller improvements (average +0.4% absolute across models). We hypothesize that HeadQA is bottlenecked by high baseline scores (∼90%), while Medec likely reflects fundamental limits in the LM's knowledge base.

**Ranking stability analysis.** To assess how structured prompting shifts relative rankings, we compute mean ranks (1 = best, 4 = worst) across all seven benchmarks. Under structured prompting, the ranking spread compresses: *o3 Mini* remains the top model but becomes less dominant (1.29 → 1.57), *Claude 3.7 Sonnet* improves and moves closer to the top (2.29 → 2.00), *GPT 4o* also improves modestly (3.14 → 3.00), while *Gemini 2.0 Flash* slightly declines (3.29 → 3.43). Rank standard deviation ($\sigma$) shows a similar but more moderate pattern: *Claude 3.7 Sonnet* (0.95 → 1.15) and *GPT 4o* (0.90 →) 1.00) become slightly more volatile, while *Gemini 2.0 Flash* stabilizes (0.76 → 0.53), and *o3 Mini* remains roughly unchanged (0.76 → 0.79). Overall, the results demonstrate that leaderboard rankings are not invariant to prompt design.

**CoT reduces sensitivity to prompt design.** We study the impact of each prompting method on the leaderboard by averaging results across LMs and benchmarks. Moving from HELM's baseline to Zero-Shot Predict yields minimal improvement (64.6% → 64.9%). In contrast, introducing CoT reasoning and moving from Zero-Shot Predict to Zero-Shot CoT results in substantial gains (64.9% → 68.5%). Interestingly, moving from Zero-Shot CoT to more sophisticated optimizers, such as BFRS and MIPROv2, does not lead to a meaningful additional improvement (68.5% → 68.6%), indicating that once CoT is introduced, LMs become less sensitive to further optimization.

| Benchmark | Prompting Method | Claude 3.7 Sonnet | Gemini 2.0 Flash | GPT 4o | o3 Mini |
|---|---|---|---|---|---|
| MedCalc-Bench | HELM Baseline | 21.0% ± 2.5 | 15.8% ± 2.3 | 18.8% ± 2.5 | 34.0% ± 3.0 |
| | Zero-Shot Predict | 20.6% ± 2.6 | 17.0% ± 2.4 | 15.7% ± 2.3 | 33.4% ± 2.9 |
| | Zero-Shot CoT | **35.3% ± 3.0 ↑** | **26.3% ± 2.7** | 26.6% ± 2.8 | 34.2% ± 3.0 |
| | BFRS | 34.1% ± 3.0 | 25.2% ± 2.7 | **27.0% ± 2.8** | **34.7% ± 3.0 ↓** |
| | MIPROv2 | 34.7% ± 3.0 | 25.4% ± 2.7 | 26.8% ± 2.8 | 34.3% ± 3.0 |
| Medec | HELM Baseline | **62.8% ± 3.9** | 59.6% ± 4.0 | 58.0% ± 3.9 | 68.7% ± 3.9 |
| | Zero-Shot Predict | 58.3% ± 3.9 | 59.3% ± 4.0 | 57.3% ± 4.0 | 68.3% ± 3.9 |
| | Zero-Shot CoT | 61.8% ± 4.0 | 59.5% ± 4.0 | 59.5% ± 4.0 | 68.2% ± 3.9 |
| | BFRS | 60.5% ± 3.9 | 59.1% ± 4.0 | 59.5% ± 4.0 | **69.2% ± 3.7** |
| | MIPROv2 | 62.5% ± 3.9 | **60.8% ± 4.0** | **59.8% ± 4.0** | 68.3% ± 3.7 |
| HeadQA | HELM Baseline | 91.2% ± 1.8 | 88.0% ± 2.1 | 90.6% ± 1.8 | 89.3% ± 1.9 |
| | Zero-Shot Predict | 88.7% ± 2.0 | 88.5% ± 2.1 | 86.4% ± 2.1 | **90.9% ± 1.8** |
| | Zero-Shot CoT | **92.2% ± 1.7** | 89.3% ± 1.9 | 90.7% ± 1.8 | 90.0% ± 1.9 |
| | BFRS | 92.0% ± 1.8 | 88.9% ± 1.9 | **91.1% ± 1.8** | 90.1% ± 1.9 |
| | MIPROv2 | **92.2% ± 1.7** | **89.5% ± 2.0** | **91.1% ± 1.8** | 89.5% ± 2.0 |
| MedBullets | HELM Baseline | 64.9% ± 5.5 | 63.0% ± 5.5 | 71.1% ± 5.2 | 81.2% ± 4.6 |
| | Zero-Shot Predict | 65.3% ± 5.5 | 64.9% ± 5.2 | 71.4% ± 5.2 | 81.5% ± 4.2 |
| | Zero-Shot CoT | 71.8% ± 5.2 | **70.8% ± 5.2** | 72.1% ± 5.2 | 81.5% ± 4.6 |
| | BFRS | 71.4% ± 5.2 | 69.5% ± 5.2 | 72.7% ± 5.2 | **82.5% ± 4.6** |
| | MIPROv2 | **72.7% ± 5.2** | 69.8% ± 5.2 | **73.4% ± 4.9** | 81.5% ± 4.6 |

Table 5: MedHELM leaderboard (medical domain) across four language models and five prompting methods. **Green** marks the "ceiling" performance for a model (best value across prompting methods). ↑ and ↓ indicate a one-step increase or decrease in leaderboard rank, respectively. Entries are reported as mean ± 95% bootstrap confidence interval. Overall, structured prompting consistently improves the robustness of benchmarks.

## 3.2 Theoretical Insight: "Why CoT Reduces Sensitivity to Prompt Design"

We first formalize the effect of CoT on prompt sensitivity. Consider a LM with parameters $\theta$, input $x$, and two prompts $p$ and $p'$ that share the same CoT interface but differ in instructions and/or demonstrations[3]. Under prompt $p$, the model samples a full reasoning path $\tau$ (CoT) and then produces a final answer $y$; i.e.,

$$P_\theta(\tau, y \mid x, p) = P_\theta(\tau \mid x, p)\, P_\theta(y \mid x, \tau, p). \tag{2}$$

The predictive answer distribution under $p$ is obtained by marginalizing over reasoning paths (self-consistency):

$$P_\theta(y \mid x, p) = \sum_\tau P_\theta(\tau \mid x, p)\, P_\theta(y \mid x, \tau, p). \tag{3}$$

Once a full reasoning path $\tau$ has been generated, the residual dependence of $y$ on the prompt is negligible:

$$P_\theta(y \mid x, \tau, p) \approx P_\theta(y \mid x, \tau, p') \approx P_\theta(y \mid x, \tau). \tag{4}$$

Because all structured prompt variants instruct the LM to output a reasoning trace, once $\tau$ is fixed, small changes in the instructions/demonstrations do not systematically change the conditional distribution over $y$:

$$p \to \tau \to y \quad \text{forms a Markov chain given } x, \tag{5}$$

---

[3]Throughout, we fix the decoding temperature and sampling strategy, so that changing $p$ only affects the textual prefix.

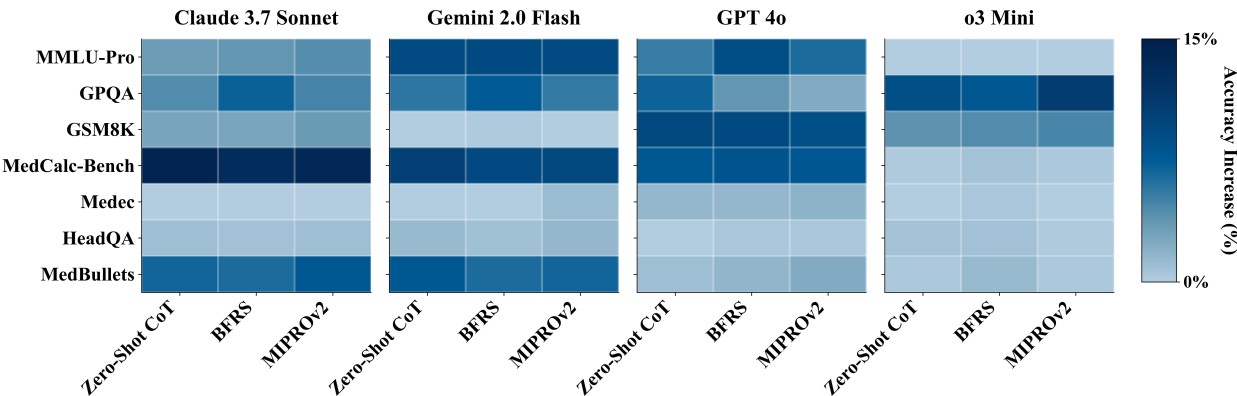

Figure 3: Heat map showing $\Delta$ (increase in accuracy) of each prompting method over HELM's baseline (light=small, dark=large). Across four models, x-axis lists prompting methods, y-axis lists benchmarks. All structured prompting methods exhibit similar improvements, while *o3 Mini* remains relatively insensitive.

i.e., $y \perp p \mid (x, \tau)$. The answer distribution $P_\theta(y \mid x, p)$ is obtained by passing the path distribution $P_\theta(\tau \mid x, p)$ through a fixed channel $P_\theta(y \mid x, \tau)$. Let $\|\cdot\|_{\mathrm{TV}}$ denote total variation distance and $D_{\mathrm{KL}}(\cdot \| \cdot)$ Kullback–Leibler divergence. Because $P_\theta(y \mid x, p)$ is the image of $P_\theta(\tau \mid x, p)$ under $\tau \mapsto y$, the data-processing inequality yields

$$\left\| P_\theta(y \mid x, p) - P_\theta(y \mid x, p') \right\|_{\mathrm{TV}} \leq \left\| P_\theta(\tau \mid x, p) - P_\theta(\tau \mid x, p') \right\|_{\mathrm{TV}}. \tag{6}$$

Applying Pinsker's inequality to the right-hand side gives

$$\left\| P_\theta(y \mid x, p) - P_\theta(y \mid x, p') \right\|_{\mathrm{TV}} \leq \sqrt{\tfrac{1}{2} D_{\mathrm{KL}}\big(P_\theta(\tau \mid x, p) \,\|\, P_\theta(\tau \mid x, p')\big)}. \tag{7}$$

Thus, the extent to which the answer distribution can change under prompt perturbations is upper-bounded by how much the CoT path distribution changes. For a given prompt $p$ and item $x$, define the decision margin

$$m(x; p) = P_\theta(y^\star \mid x, p) - \max_{y \neq y^\star} P_\theta(y \mid x, p), \qquad y^\star = \arg\max_y P_\theta(y \mid x, p). \tag{8}$$

We now state a pointwise decision-stability result. Fix $x$ and prompts $p, p'$. If

$$\left\| P_\theta(y \mid x, p) - P_\theta(y \mid x, p') \right\|_{\mathrm{TV}} < \tfrac{1}{2} m(x; p), \tag{9}$$

then the prediction is invariant:

$$\arg\max_y P_\theta(y \mid x, p') = \arg\max_y P_\theta(y \mid x, p). \tag{10}$$

Moreover, a sufficient condition is

$$D_{\mathrm{KL}}\big(P_\theta(\tau \mid x, p) \,\|\, P_\theta(\tau \mid x, p')\big) \leq \kappa \quad \text{and} \quad m(x; p) \geq 2\varepsilon \implies \sqrt{\kappa/2} < \varepsilon \Rightarrow \textit{equation } 10. \tag{11}$$

Condition 9 implies that the probability mass on the top-class $y^\star$ cannot be reduced by more than $\tfrac{1}{2} m(x; p)$, while the mass on any competitor cannot be increased by more than the same amount. Hence no competitor can overtake $y^\star$, giving 10. Inequality 11 combines the TV bound in 7 with the margin condition 9.

In CoT decoding, $P_\theta(y \mid x, p)$ can be viewed as a marginalization over possible reasoning paths, which typically enlarges the margin $m(x; p)$ compared to direct (non-CoT) decoding. At the same time, structured prompting methods mainly alter instructions and few-shot examples while preserving the CoT interface, so they primarily act by *reweighting* $P_\theta(\tau \mid x, p)$ rather than changing the conditional channel $P_\theta(y \mid x, \tau)$. Once CoT is enabled, the effective KL divergence between path distributions under different structured prompts is small enough that equation 11 holds for most items, and further optimization rarely flips decisions except on near-tied examples. Empirically, this is reflected in our results: moving from non-CoT to Zero-Shot CoT yields majority gains, while more aggressive optimizers (BFRS, MIPROv2) produce marginal improvements.

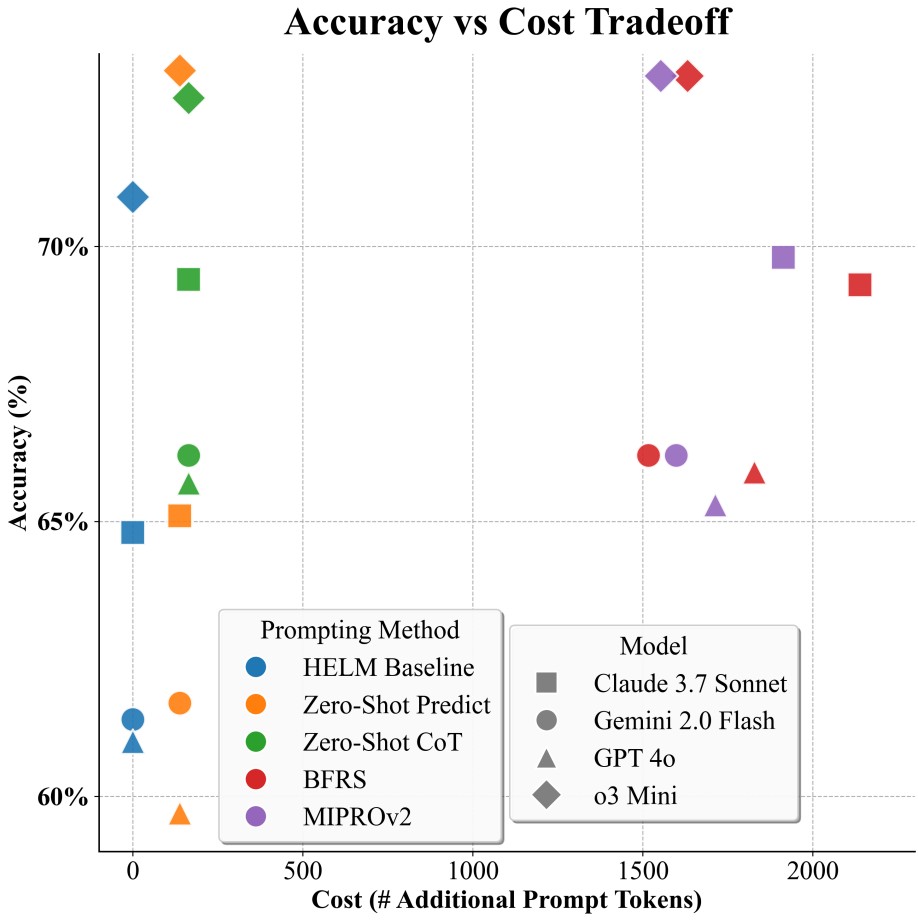

Figure 4: Accuracy vs cost tradeoff across prompting methods. Each point represents a model-prompt pair, with x-axis showing additional prompt tokens (relative to HELM baseline) and y-axis showing macro-averaged accuracy across benchmarks. Overall, Zero-Shot CoT is the most cost-effective structured prompting method.

### 3.3 Computational Cost Analysis

We evaluate the inference-time computational cost across prompting methods by examining token usage. BFRS and MIPROv2 optimizations are one-time expenses amortized over future runs: DSPy's documentation reports that optimization runs range from a few cents to tens of dollars depending on the configuration,[4]. Our optimization costs match DSPy's reported range. We therefore focus our analysis on inference-time tokens.

In our setup, the input (e.g., question, patient note) is identical across prompting methods, and outputs are capped at <200 tokens. As a result, differences in inference cost arise almost entirely from the *prompt prefix* (the instructions and demonstrations prepended to the input). Hence, we quantify the number of *additional prompt tokens* relative to the HELM baseline instruction, capturing each prompting method's overhead.

DSPy introduces a lightweight structured prompt template across all methods, resulting in 138 additional tokens for Zero-Shot Predict and 164 tokens for Zero-Shot CoT, which further includes a brief reasoning header. In contrast, BFRS and MIPROv2 insert task-specific demonstrations, producing much larger prompts: averaged across LMs and benchmarks, BFRS adds 1,779 tokens per query and MIPROv2 adds 1,694.

Figure 4 shows the resulting tradeoff. Few-shot optimizers reach high ceilings but require the largest token budgets. Zero-Shot CoT captures most of these gains while using minimal additional prompt tokens, making Zero-Shot CoT the most cost-effective structured prompting method in our study.

---

[4]DSPy Optimizer Costs: `https://dspy.ai/learn/optimization/optimizers/`.

## 4 Related Work

**Holistic benchmarking.** The General Language Understanding Evaluation (GLUE) (Wang et al., 2018) benchmark was one of the first multi-task evaluation frameworks, aggregating nine distinct language understanding tasks. Benchmarks of increasing scale followed: (i) Measuring Massive Multitask Language Understanding (MMLU) (Hendrycks et al., 2020), including 57 tasks spanning STEM, humanities, social sciences, and (ii) Beyond the Imitation Game (BIG-Bench) (Srivastava et al., 2023), with 204 diverse tasks. The HELM framework is an established standard, designed for transparent, reproducible, and multi-metric evaluation of model capabilities (Liang et al., 2022). However, these benchmarks are typically evaluated using static prompts. Liang et al. (2022) note they opt for simple, generic prompts to orient development "towards generic language interfaces" that do not require "model-specific incantations". This reliance on fixed prompts, however, risks the underestimation of the true capabilities of LMs. Srivastava et al. (2023); Suzgun et al. (2023) conclude that standard few-shot prompting substantially underestimates the capabilities of LMs.

**Prompting methods.** The discovery of in-context learning (Brown et al., 2020), where models learn from n-shot demonstrations, and the breakthrough of chain-of-thought (CoT) prompting (Wei et al., 2022) established the important role of prompt design in model performance. Complex, manually-composed strategies like Medprompt (Nori et al., 2023), which combine few-shot selection, CoT, and ensembling, demonstrate that a LM's performance ceiling often lies higher than with the use of static prompts. Because manual prompt engineering is impractical for systematically approximating this ceiling, researchers often frame prompt design as a formal "optimization problem", leading to the field of APO. Early APO methods include generation-and-selection, such as Automatic Prompt Engineer (APE) (Zhou et al., 2022), which uses an LM to propose candidate instructions and a separate scoring function to select the best one. Subsequent systems expanded this search paradigm (Wang et al., 2023; Yang et al., 2023; Singla et al., 2024). These methods often outperform zero-shot or manually engineered prompts on a variety of general tasks. In the LM-as-Optimizer paradigm, an LM is instructed to iteratively refine prompts by showing it a trajectory of previously evaluated candidates and their scores. Other approaches have employed evolutionary search, like Promptbreeder (Fernando et al., 2024), which treats prompts as "genes" and evolves a population of instructions over generations using a LM to perform mutation. The DSPy framework (Khattab et al., 2023) generalizes these methods, providing a programming model that compiles declarative, multi-stage pipelines.

## 5 Limitations

First, we focus on widely used frontier LMs rather than open-source models. While this choice highlights that even strong models remain sensitive to prompt design, it limits the generality of our findings because frontier LMs differ in training data transparency, accessibility, and reproducibility compared to open-source models. Second, our benchmarks primarily involve multiple-choice and short-form reasoning tasks, and results may not generalize to open-ended generation tasks. Third, we evaluate a subset of structured prompting methods from the DSPy family; alternative frameworks could yield higher ceilings. However, our goal is not to identify the optimal prompting method, but to demonstrate that fixed-prompt (without CoT) leaderboard evaluations can systematically underestimate LM performance and often distort model comparisons and rankings.

## 6 Conclusion

By integrating DSPy with HELM, we empirically approximate LM performance ceilings, obtaining more representative estimates. Our results show that structured prompting can materially alter benchmark conclusions, shifting relative LM ordering and improving robustness by reducing sensitivity to arbitrary prompt choices. Sensitivity is heterogeneous: reasoning LMs show marginal gains, whereas some benchmarks for non-reasoning LMs benefit more, and gains are largely agnostic to the particular structured prompting method. The key driver of improvement is the transition from the baseline prompt to *any* CoT variant, with Zero-Shot CoT providing the most cost-efficient instantiation. Future public leaderboards should report performance under multiple structured prompting methods, enabling practitioners to assess achievable performance across prompting styles and make more informed deployment decisions. Together, we show that scalable and automated performance-ceiling approximation enables more robust, decision-useful benchmarks.

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

## Appendix

## A Bootstrap Few-Shot with Random Search (BFRS)

**Problem setup.** Let $\Phi$ be a LM program with $m$ modules $\{\Phi_i\}_{i=1}^m$, each parameterized by (i) an instruction string $I_i \in \mathcal{I}_i$ and (ii) an ordered list of $K_i$ few-shot demonstrations $S_i = [(u_i^{(1)}, v_i^{(1)}), \ldots, (u_i^{(K_i)}, v_i^{(K_i)})]$ drawn from a module-specific pool $\mathcal{B}_i$. Write the prompt configuration as $\mathbf{v} = (I_1, S_1, \ldots, I_m, S_m) \in \mathcal{V}$, and define the end-to-end score on $(x, y)$ as $\mu(\Phi_{\mathbf{v}}(x), y) \in [0, 1]$. Given train/validation splits $D_{\text{tr}}, D_{\text{val}}$, the objective is

$$J(\mathbf{v}) \;=\; \frac{1}{|D_{\text{val}}|} \sum_{(x,y) \in D_{\text{val}}} \mu\big(\Phi_{\mathbf{v}}(x), y\big). \tag{12}$$

**Bootstrapping demonstration pools.** BFRS constructs candidate demo pools $\{\mathcal{B}_i\}_{i=1}^m$ via rejection sampling with a seed program $\Phi_{\text{seed}}$ (typically zero-shot):

$$\mathcal{B}_i \;=\; \Big\{ (u_i(x), \, \hat{v}_i(x)) \;\Big|\; (x,y) \in D_{\text{tr}}, \; \mu\big(\Phi_{\text{seed}}(x), y\big) \geq \tau, \; \hat{v}_i(x) = \Phi_{\text{seed}}^{(i)}\big(u_i(x)\big) \Big\}, \tag{13}$$

where $u_i(x)$ denotes input to module $i$ induced by running $\Phi_{\text{seed}}$ on $x$, and $\tau \in [0, 1]$ is an acceptance threshold.

**Random search over few-shots.** With instructions fixed at $I_i = I_i^{\text{seed}}$, BFRS draws $R$ candidates

$$S_i^{(r)} \sim \text{SampleK}\big(\mathcal{B}_i, K_i\big) \quad \text{and} \quad \mathbf{v}^{(r)} = (I_1^{\text{seed}}, S_1^{(r)}, \ldots, I_m^{\text{seed}}, S_m^{(r)}), \tag{14}$$

evaluates $\widehat{J}_B(\mathbf{v}^{(r)})$ on a size-$B$ minibatch of $D_{\text{val}}$,

$$\widehat{J}_B(\mathbf{v}) \;=\; \frac{1}{B} \sum_{(x,y) \in \mathcal{B} \subset D_{\text{val}}} \mu\big(\Phi_{\mathbf{v}}(x), y\big), \tag{15}$$

and returns the best $\mathbf{v}^\star \in \arg\max_{r \in [R]} \widehat{J}_B(\mathbf{v}^{(r)})$; optionally, $\mathbf{v}^\star$ is re-scored on the full $D_{\text{val}}$. Since $\mu \in [0, 1]$, Hoeffding implies $|\widehat{J}_B(\mathbf{v}) - J(\mathbf{v})| \leq \sqrt{\frac{\ln(2/\delta)}{2B}}$ with prob. $\geq 1 - \delta$.

## B MIPROv2

**Search space and objective.** As in §A, $\mathbf{v} = (I_1, S_1, \ldots, I_m, S_m)$ parameterizes $\Phi$ and $J(\mathbf{v})$ is defined in equation 12. MIPROv2 *jointly* searches over instructions and few-shot demos by (a) bootstrapping demo candidates $\{\mathcal{B}_i\}$, (b) proposing instruction candidates $\{\mathcal{I}_i\}$ via a *proposer LM*, and (c) using Bayesian Optimization (BO) with a Tree-structured Parzen Estimator (TPE) surrogate to choose $\mathbf{v}$.

**Initialization (proposal sets).** For each module $i$, construct

$$\underbrace{\mathcal{B}_i}_{\text{demo sets}} \quad \text{by bootstrapping as in equation 13,} \qquad \underbrace{\mathcal{I}_i}_{\text{instruction set}} \;=\; \big\{ I_i^{(1)}, \ldots, I_i^{(T_i)} \big\},$$

where $I_i^{(t)} \sim q_i(\cdot \mid \text{ctx}_i)$ are sampled by the proposer LM given context $\text{ctx}_i$.

**Bayesian surrogate via TPE.** Maintain a history $\mathcal{H}_t = \{(\mathbf{v}^{(s)}, y^{(s)})\}_{s=1}^t$ of tried configurations and noisy scores $y^{(s)} = \widehat{J}_{B_s}(\mathbf{v}^{(s)})$ from minibatches. Let $y^\star$ be the $\gamma$-quantile of $\{y^{(s)}\}_{s=1}^t$ (e.g., $\gamma = 0.2$). TPE models

$$\ell(\mathbf{v}) \;=\; p(\mathbf{v} \mid y < y^\star), \qquad g(\mathbf{v}) \;=\; p(\mathbf{v} \mid y \geq y^\star), \tag{16}$$

and proposes $\mathbf{v}$ to maximize the ratio $\ell(\mathbf{v})/g(\mathbf{v})$ (improvement proxy). With categorical choices per module, TPE factorizes $\mathbf{v}$ across modules/slots and estimates equation 16 from $\mathcal{H}_t$ by smoothed frequency models.

**Noisy validation and escalation.** Each candidate $\mathbf{v}$ is scored on a minibatch $\mathcal{B}$ of size $B$ by equation 15. Every $E$ trials, the current top-$K$ candidates (by posterior mean or running average) are *escalated* to full-$D_{\text{val}}$ evaluation; the best full-eval configuration $\mathbf{v}^\dagger$ to date is tracked and returned at the end. Concentration of $\widehat{J}_B$ to $J$ is controlled by $B$ (Hoeffding bound as above).

