# OpenReview forum: "Structured Prompting Enables More Robust Evaluation of Language Models"
_TMLR — Rejected by TMLR_

### Review · Reviewer_JhgW · 2025-12-21

**Summary Of Contributions:**

This paper argues that fixed-prompt benchmarking (specifically HELM’s commonly reported zero-shot, no-CoT baseline) can substantially underestimate a language model’s achievable performance, and can distort cross-model comparisons. The authors integrate DSPy structured prompting into HELM and evaluate four prompting configurations (HELM baseline, DSPy zero-shot predict, DSPy zero-shot CoT, BFRS, MIPROv2) on four frontier LMs across seven HELM benchmarks spanning general + medical tasks. They report that structured prompting, especially adding CoT, improves macro-averaged accuracy (around +4-5% absolute for non-reasoning models) and reduces across-benchmark variance, and that leaderboard rankings can flip under the "ceiling" prompt (best prompt among tested methods). They also provide a token-cost analysis and a short theoretical argument for why CoT reduces prompt sensitivity.

Key strengths
1. Concrete, actionable message: HELM-style fixed prompts can change conclusions, including ranks, and this is demonstrated on multiple well-known benchmarks and multiple frontier APIs.
2. Clear experimental structure: same tasks, multiple prompting methods, and explicit reporting of "ceiling - baseline" deltas.
3. Practical cost discussion (prompt token overhead) that is relevant to deployment tradeoffs.

Key weaknesses
1. The central "robust evaluation" framing is not convincingly supported: the paper mostly shows that "adding CoT increases accuracy," which is already well known, while the stronger claim (structured prompting yields more decision-useful, robust benchmarking) is not rigorously established.
2. The theoretical section is not aligned with the actual experimental regime (they run deterministic decoding at temperature = 0, yet the theory relies on stochastic path distributions and marginalization). This reads as post-hoc and insufficiently justified.
3. Methodological rigor is weaker than it should be for a benchmarking paper: single runs, limited model set (4), limited task variety (mostly multiple-choice / short-form), and unclear robustness to random seeds or optimizer overfitting to validation.

**Additional Comments:**

1. The main empirical takeaway appears to be: most gains come from enabling CoT, while more sophisticated optimizers add little beyond that (the paper also states this). This makes the "structured prompting" framing feel overstated; the title and claims should be tightened unless stronger evidence is added.
2. Reporting "macro-average ± sigma across benchmarks" is an unusual and potentially misleading summary statistic; I would prefer per-benchmark prompt-sensitivity analyses and/or aggregate metrics that are explicitly tied to robustness.
3. If the authors can address the critical methodological gaps (robustness definition, theory alignment, selection bias/overfitting, stronger baselines), the paper could become a solid evaluation methodology contribution. As submitted, I do not think the evidence matches the strength of the claims.

**Audience:**

Yes

**Audience Explanation:**

Benchmarking methodology and evaluation validity are highly relevant to TMLR. The paper provides a practical integration of DSPy-style structured prompting with HELM, highlighting a real issue: fixed prompts can lead to misleading model comparisons and an underestimation of achievable performance. Even if the main empirical story reduces to "CoT matters," the intCritical (required for acceptance)

Define and measure “robustness” correctly. Add direct prompt sensitivity experiments: for each benchmark, perturb prompts (paraphrases, formatting changes, different CoT headers, different demo orders) and report variance in performance. Across-benchmark sigma is not sufficient. Integration and systematic comparison across multiple HELM tasks is still of interest to the evaluation and reliability audience.

**Broader Impact Concerns:**

1. Leaderboard gaming: The paper encourages optimizing prompts for benchmarks and reporting "ceilings." This can further incentivize benchmark overfitting and reduce the value of leaderboards as measures of real-world performance unless strict protocols are adopted (e.g., restricted prompt search budgets, standardized reporting of prompt families). This should be explicitly discussed as a potential negative externality.
2. Medical prompting risks: Some prompts use high-stakes framing ("busy emergency room," "patient’s life may depend on your accuracy"). If reused outside benchmarking, this can encourage overconfident outputs. The broader impact section should caution against deploying such prompting patterns in clinical settings without safeguards.

**Claims And Evidence:**

No

**Claims Explanation:**

Some narrower claims are supported: the tables do show that adding CoT (and related structured prompting) increases accuracy relative to HELM’s no-CoT baseline and can change ranks on some benchmarks.

However, several key claims are not convincingly supported to a TMLR standard:
1. "Robust evaluation" is not properly operationalized. The authors treat reduced across-benchmark standard deviation as 'robustness,' but that is not a direct measure of prompt sensitivity or evaluation stability; it can drop simply because accuracy rises on low-performing tasks, compressing scores, without demonstrating reduced sensitivity to prompting within each benchmark.
2. Ceiling approximation is weakly justified. The "ceiling" is defined as the best among a small set of prompt families (mostly CoT variants). There is no evidence that this approximates a meaningful upper bound, nor that it correlates with stronger prompt search or expert tuning. "Best of five" is not a ceiling; it is a limited sweep.
3. The theory section is largely disconnected from experiments. The analysis assumes distributions over reasoning paths and invokes self-consistency-style marginalization, but the experiments use deterministic decoding (temperature = 0) and do not actually sample multiple reasoning paths. As written, the theory does not explain the reported outcomes under the stated setup.
4. Ranking flip claims are overemphasized without sufficient statistical care. While bootstrap CIs are provided for some tables, "flips" can occur with overlapping intervals and small differences, and the paper does not present a stability analysis of rank under resampling or across repeated runs / alternative evaluation seeds.

Because the paper's stronger conclusions (robust, decision-useful benchmarking via scalable ceiling approximation) are not backed by appropriately targeted evidence, I do not find the claims convincingly supported.

**Requested Changes:**

Critical (required for acceptance)

1. Define and measure "robustness" correctly. Add direct prompt sensitivity experiments: for each benchmark, perturb prompts (paraphrases, formatting changes, different CoT headers, different demo orders) and report variance in performance. Across-benchmark sigma is not sufficient.
2. Fix the theory-experiment mismatch. Either (a) run experiments that match the theory (nonzero temperature, multiple sampled CoTs, explicit marginalization/self-consistency), or (b) rewrite the theory to address deterministic decoding and clearly state assumptions. Current theory is not credible as an explanation.
3. Strengthen the "ceiling" claim. Demonstrate that your ceiling approximation is meaningful by comparing against stronger baselines:
  (i) A simple manually designed CoT baseline in HELM (same token budget as your Zero-Shot CoT).
  (ii) A prompt-budget-matched baseline (equalize prompt tokens across methods).
  (iii) Potentially an external APO method not in DSPy (even one representative alternative) to show DSPy is not cherry-picked.
4. Address overfitting/selection bias in prompt optimization. Provide details of the number of optimization trials and show that gains hold on truly held-out test sets without repeated selection effects. Ideally, include multiple random seeds for train/val splits and show stability. Single deterministic runs are not enough for an evaluation paper making robustness claims.
5. Broaden model coverage beyond frontier APIs. Include at least a few strong open-weight models to improve reproducibility and generality (the paper currently acknowledges this limitation, but it is still too constraining for the paper’s claims).

Non-critical (would strengthen):
1. Expand task diversity to include open-ended generation and longer-context tasks, where prompt structure often has different effects than multiple-choice.
2. More rigorous cost analysis: report total optimization cost (tokens and dollars) rather than citing "DSPy documentation range," and present amortization assumptions explicitly.
3. Rank stability analysis: compute rank distributions under bootstrap resampling and (if possible) repeated runs, not just point ranks.
4. Ablate the structured format vs CoT content: show whether the improvement comes from "REASONING/OUTPUT fields" formatting, from CoT instruction alone, or from other DSPy template choices.

---

> ### Author Response · Authors · 2026-01-21
> **Rebuttal by Authors**
>
> We thank the reviewer for their careful reading of the paper and for the detailed, thoughtful feedback. We appreciate their recognition of the paper’s concrete empirical message, clear experimental structure, and practical relevance. We agree that clarifying and strengthening the framing around robustness, theoretical alignment, and statistical rigor is important, and we address these points in detail below.
>
> ---
>
> **Critical (required for acceptance)**
>
> **R3.C1**: *Define and measure "robustness" correctly. Add direct prompt sensitivity experiments: for each benchmark, perturb prompts (paraphrases, formatting changes, different CoT headers, different demo orders) and report variance in performance. Across-benchmark sigma is not sufficient.*
>
> **Author Response**: We thank the reviewer for this thoughtful critique and apologize for the ambiguity in our original use of the term robustness. We agree that our earlier wording conflated several distinct concepts, and we will revise the manuscript to clarify definitions, claims, and evidence accordingly.
>
> *Clarification regarding “Robustness”*
>
> In this work, we use evaluation robustness to mean stability of reported performance estimates and rankings with respect to reasonable prompt choices, rather than variance reduction or uniform performance improvement. Under this definition, an evaluation is robust if its conclusions (headline performance and relative rankings) do not hinge on an arbitrary single prompt, but instead approximate a model’s attainable capability stably.
>
> We acknowledge that the confidence intervals and across-benchmark standard deviations reported in Tables 3-5 in the paper do not demonstrate robustness. Their purpose is solely to assess whether observed performance differences are meaningful rather than due to noise. As the reviewer correctly notes, the fact that these intervals remain similar across prompting methods indicates that the observed gains are not driven by increased variance; this supports statistical significance, not robustness. These findings are further supported by paired statistical significance tests, described in detail later, which show that the aggregate performance gaps induced by prompt choice are statistically significant at the instance level and unlikely to arise by chance. We regret the confusion and will revise table captions and surrounding text to make this distinction explicit.
>
> Our claim rests on three observations:
> 1. *Large performance shifts when moving from non-CoT to CoT prompts indicate the relative instability of fixed, non-CoT evaluation*: A simple, generic CoT instruction, without task-specific optimization, produces substantial performance changes, showing that fixed non-CoT prompts can underrepresent model capability and yield rankings that are sensitive to incidental prompt choice. In contrast, once a CoT interface is introduced, performance stabilizes across different structured prompting variants, and comparable prompt changes no longer induce large shifts.
> 2. *Convergence within CoT-based prompting indicates increased stability*: Once CoT is introduced, performance estimates stabilize: Zero-Shot CoT, BFRS, and MIPROv2, despite substantial differences in structure, yield similar results (e.g., 68.5% → 68.6% on average). The absence of large swings within CoT-based methods contrasts sharply with the large jump from non-CoT to CoT, suggesting reduced sensitivity relative to baseline, under the prompt families we tested.
> 3. *Model-specific evidence (e.g., o3-mini) reinforces this pattern*: For the reasoning-optimized model o3-mini, performance deltas across all prompting strategies (CoT or non-CoT) are consistently small, indicating less sensitivity to prompt formulation. This further supports the view that CoT-based evaluation can yield more stable estimates of model capability.

---

> > ### Author Response · Authors · 2026-01-21
> > **Rebuttal by Authors (Continued)**
> >
> > **R3.C1 (CONTINUED)**
> >
> > **Author Response**:
> >
> > *New Experiment 1: Statistical Significance Testing of Performance Differences*
> >
> > To ensure that observed performance gaps are not driven by noise or small subsets of instances, we added paired statistical testing throughout the evaluation. Specifically, we apply an exact two-sided McNemar test on paired binary outcomes (correct/incorrect) for each prompting method against the HELM baseline, with Benjamini–Hochberg FDR correction for multiple comparisons.
> >
> > In addition to per-benchmark tests, we introduce a pooled significance analysis that aggregates paired comparisons across all benchmark instances for each model. Under this pooled analysis, structured prompting methods (Zero-Shot CoT, BFRS, and MIPROv2) significantly outperform the HELM baseline for all four frontier models (p < 0.05 after FDR correction). This pooled result is critical for our central claim: it demonstrates that the performance gaps induced by prompt choice reflect consistent per-instance improvements across benchmarks, rather than chance effects.
> >
> > At the benchmark level, we observe significance primarily where performance differences are substantial; conversely, on benchmarks with minimal differences (e.g., Medec, HeadQA), McNemar’s test does not detect significance, which is expected and supports a conservative interpretation. We further observe that the reasoning-specialized model o3-mini exhibits fewer significant changes across prompts, consistent with its reduced prompt sensitivity.
> >
> > *New Experiment 2: Controlled Prompt Variation Analysis*
> >
> > While we believe broad prompt-perturbation studies are an important direction for future work, we conducted a controlled prompt variation analysis to isolate the effect of enabling CoT under otherwise matched prompt content. We compared:
> > 1. Non-CoT Variants: (i) Zero-Shot HELM Baseline, (ii) In-Context Demonstrations without CoT, (iii) Optimized Instructions with Demonstrations but without CoT
> > 2. CoT Structured Variants: (i) Zero-Shot CoT, (ii) BFRS, (iii) MIPROv2
> >
> > Importantly, the Non-CoT variants were constructed to match the CoT-based prompts in instructions and demonstrations, differing only in the absence of the DSPy CoT interface. This controlled experiment isolates the impact of CoT itself, allowing us to assess prompt sensitivity under comparable prompt complexity.
> >
> > For MedCalc-Bench, we conducted a focused subset analysis using two representative models, a widely used frontier model (GPT-4o) and a smaller open-source model (Qwen3-4B), to examine sensitivity to structured prompt variation across model scales. For Qwen3-4B, Non-CoT prompt variants yield a mean±stdev accuracy of 8.7% ± 3.5%, whereas CoT-based structured prompts yield 21.7% ± 1.0%, indicating both substantially higher performance and markedly lower variance under CoT. Similarly, for GPT-4o, Non-CoT variants achieve 19.9% ± 1.0%, while CoT-based prompts reach 26.8% ± 0.2%, again exhibiting reduced sensitivity relative to baseline, under the prompt families we tested. While limited in scope, we hope this analysis provides more concrete support for our claim. Accordingly, we will revise the introduction, discussion, and table captions to reflect our more precise claim and explicitly frame broader prompt sensitivity analysis as an important direction for future work.
> >
> > ---
> >
> > **R3.C2**: *Fix the theory-experiment mismatch. Either (a) run experiments that match the theory (nonzero temperature, multiple sampled CoTs, explicit marginalization/self-consistency), or (b) rewrite the theory to address deterministic decoding and clearly state assumptions. Current theory is not credible.*
> >
> > **Author Response**: We agree that the original theory implicitly assumed stochastic decoding, while some prior analyses of CoT consider nonzero temperature sampling and explicit marginalization. To resolve this mismatch, we will revise the theory to explicitly target the deterministic (temperature-zero) decoding regime used in all of our experiments, rather than introducing sampling-based assumptions that are not present in our empirical setup. In the revised analysis, the theory will be framed as a conceptual explanation that provides sufficient conditions for prediction stability under prompt variation, rather than as an empirical model of stochastic generation. All results are stated under deterministic decoding, where predictions are fixed functions of the input and prompt. Concretely, the revised theory will (i) introduce a formal deterministic setup and explicit assumptions, (ii) prove a data-processing inequality showing that sensitivity of the final prediction is upper-bounded by sensitivity in the induced reasoning trace, and (iii) establish a margin-based decision-stability theorem guaranteeing invariance of the argmax prediction under bounded prompt variations. We will further (iv) derive sufficient conditions for this stability in terms of divergence between reasoning-path distributions.

---

> > > ### Author Response · Authors · 2026-01-21
> > > **Rebuttal by Authors (Continued)**
> > >
> > > **R3.C3**: *Strengthen the "ceiling" claim. There is no evidence that this approximates a meaningful upper bound. Demonstrate that your ceiling approximation is meaningful by comparing against stronger baselines: (i) A simple manually designed CoT baseline in HELM (same token budget as your Zero-Shot CoT). (ii) A prompt-budget-matched baseline (equalize prompt tokens across methods). (iii) Potentially an external APO method not in DSPy (even one representative alternative) to show DSPy is not cherry-picked.*
> > >
> > > **Author Response**: We thank the reviewer for raising this point and agree that our original use of the term ceiling was imprecise. Our intent is not to claim that we approximate a global or task-optimal upper bound on model performance, nor that our results constitute a comprehensive prompt search. We will revise the manuscript (including the abstract and discussion) to clarify this terminology throughout. In the revised version, what we previously referred to as a “ceiling” will be more precisely described as the best-performing prompt among a small, standardized set of reasonable and widely used prompting strategies.
> > >
> > > Furthermore, we would like to clarify our prompting method selection criteria. First, our goal is not to benchmark or advocate for any specific evaluation framework (e.g., HELM) or optimization framework (e.g., DSPy). Rather, we claim that leaderboard outcomes are sensitive to prompt choice, and HELM and DSPy are used because they are widely adopted and provide a transparent, reproducible way to vary prompting strategies within an established benchmark. Second, DSPy is not a single optimizer but a framework that enables movement along a spectrum of commonly used prompting regimes, from minimal adaptation (Zero-Shot Predict, Zero-Shot CoT) to higher adaptation (BFRS, MIPROv2), making them reasonable prompt choices for studying effects on reported results, rather than for exhaustively maximizing performance.
> > >
> > > Regarding the reviewer’s suggested comparisons, we agree that these are valuable for studying prompt efficiency. However, we believe our central claim is framework-agnostic: fixed-prompt benchmarks can substantially underrepresent model capability, because there exist other standard, easily accessible prompts that yield materially higher performance and altered rankings. Identifying a true performance upper bound, we believe, is orthogonal to this goal and outside the scope of our work; we note comprehensive prompt search as an important direction for future research, as explored in LangProBe [1].
> > >
> > > However, we see value in exploring an additional non-DSPy-based APO to address the concern that only a subset of DSPy-based methods was evaluated. We ran GEPA [2], a recent prompt optimization algorithm that has been shown to outperform reinforcement-learning-based approaches in prior work. Importantly, we use the standalone GEPA implementation, not a DSPy instantiation, to ensure that our findings are not an artifact of DSPy itself. On MMLU-Pro, GEPA achieves performance comparable to existing DSPy optimizers across models. These results (Table 1) reinforce that the precise choice of optimizer is less important than the fact that alternative prompting strategies can yield performance substantially higher than currently reported leaderboard baselines. In the revised paper, we will explicitly add both (i) a discussion clarifying the rationale for method selection and (ii) the new GEPA results (including the MMLU-Pro table).
> > >
> > > [1] Tan, Shangyin, et al. "LangProBe: a Language Programs Benchmark." EMNLP (2025).
> > >
> > > [2] Agrawal, Lakshya A., et al. "Gepa: Reflective prompt evolution can outperform reinforcement learning." arXiv preprint arXiv:2507.19457 (2025).
> > >
> > > Table 1: Average accuracy (%) across 6 prompting methods and 6 language models, including **GEPA** optimizer.
> > >
> > > | Prompting Method      | Claude 3.7 | Gemini 2.0 | GPT-4o | o3 Mini | Llama3.3 | Qwen3 |
> > > |-----------------------|------------|------------|--------|---------|----------|-------|
> > > | HELM Baseline         | 76.3%      | 66.1%      | 62.2%  | 77.1%   | 64.7%    | 44.9% |
> > > | Zero-Shot Predict    | 77.7%      | 70.3%      | 60.7%  | 78.4%   | 62.7%    | 41.7% |
> > > | Zero-Shot CoT        | 79.7%      | 75.3%      | 67.6%  | 76.2%   | 68.5%    | 66.2% |
> > > | BFRS                 | 80.1%      | 75.4%      | 71.1%  | 76.5%   | 68.5%    | 68.6% |
> > > | MIPROv2              | 80.6%      | 75.3%      | 68.7%  | 76.1%   | 56.5%    | 68.6% |
> > > | **GEPA**              | **79.0%**  | **69.5%**  | **73.2%** | **77.6%** | **63.4%** | **55.1%** |

---

> > > > ### Author Response · Authors · 2026-01-21
> > > > **Rebuttal by Authors (Continued)**
> > > >
> > > > **R3.C4**: *Address overfitting/selection bias in prompt optimization. Provide details of the number of optimization trials and show that gains hold on truly held-out test sets without repeated selection effects. Ideally, include multiple random seeds for train/val splits and show stability. Single deterministic runs are not enough for an evaluation paper making robustness claims.*
> > > >
> > > > **Author Response**: We thank the reviewer for raising this important point. To address the concern, we will clarify the optimization protocol, report the number of trials, and sharpen the scope of our claims in the implementation details and discussion sections of the revised manuscript.
> > > >
> > > > *Data separation and absence of test leakage*
> > > >
> > > > For all optimized methods (BFRS and MIPROv2), prompt selection follows DSPy’s standard data separation. Demonstrations are bootstrapped exclusively from the training split, and candidate prompts are evaluated only on a disjoint held-out validation split drawn from the original training partition (default 90/10 with a fixed seed). Crucially, the HELM leaderboard test set is never used during optimization, and HELM computes all final reported results on the held-out evaluation split. As a result, there is no repeated selection on test data.
> > > >
> > > > *Optimization budget and determinism*
> > > >
> > > > We will now explicitly report optimization budgets in the revised manuscript: BFRS uses 16 trials, and MIPROv2 uses 13 trials, with capped demonstrations (K ≤ 3 per module). All runs use deterministic decoding (temperature = 0), which matches HELM’s evaluation protocol and eliminates stochastic sampling variance. Under this setup, variability arises only from prompt structure, not from generation randomness.
> > > >
> > > > *Multiple seeds and stability*
> > > >
> > > > We agree that multi-seed optimization and nested validation are important when the goal is to estimate optimal prompt performance of a specific optimizer. However, our goal is to particularly show that fixed-prompt benchmarking can underrepresent model capability even under minimal, deterministic prompt variation. We therefore intentionally mirror HELM’s single-run, deterministic evaluation protocol to isolate the effect of prompt choice itself. We will explicitly note broader stability analyses across seeds at both train/inference time as an important future direction in the revised manuscript.

---

> > > > > ### Author Response · Authors · 2026-01-21
> > > > > **Rebuttal by Authors (Continued)**
> > > > >
> > > > > **R3.C5**: *Broaden model coverage beyond frontier APIs. Include at least a few strong open-weight models to improve reproducibility and generality (the paper currently acknowledges this limitation, but it is still too constraining for the paper’s claims).*
> > > > >
> > > > > **Author Response**: We thank the reviewer for highlighting the importance of generalizability beyond frontier models. To clarify, our submission focuses on frontier LMs intentionally: the goal was to demonstrate that even state-of-the-art models can exhibit sensitivity to prompt design, implying that fixed-prompt leaderboards can underrepresent capability via altered performance estimates and rankings. That said, we agree that explicitly validating this behavior across open-source models would strengthen the analysis.
> > > > > In response, we have expanded the experimental section to include two open-source models: Llama-3.3-70B-Instruct and Qwen3-4B-Instruct-2507, which represent complementary points in the open-source landscape, one large, frequently reported model on public benchmarks, and one smaller model known for strong performance relative to its parameter count. The new experiments  (Table 2 and 3) are consistent with our main findings and support the generality of our conclusions.
> > > > >
> > > > > 1. Structured prompting methods consistently outperform HELM’s fixed baseline, mirroring the trends observed for frontier models.
> > > > > 2. Across Llama-3.3-70B and Qwen3-4B, introducing chain-of-thought (Zero-Shot CoT) yields the majority of performance gains, while more sophisticated optimizers (BFRS, MIPROv2) provide limited additional benefit.
> > > > > 3. On MedCalc-Bench, when evaluated at ceiling (best among evaluated prompting strategies), Qwen3-4B not only closes the gap but slightly outperforms Llama-3.3-70B, despite being roughly 18x smaller. This further reinforces our central claim: fixed-prompt leaderboards can underrepresent performance, and reported rankings may reflect prompt choice rather than capability.
> > > > >
> > > > > Table 2: Average accuracy (%) across 5 prompting methods and 7 benchmarks for **Llama3.3-70B**.
> > > > >
> > > > > | Prompting Method   | MMLU-Pro | GPQA | GSM8K | MedCalc | Medec | HeadQA | MedBullets |
> > > > > |--------------------|----------|------|-------|---------|-------|--------|------------|
> > > > > | HELM Baseline      | 64.7%    | 57.0%| 85.1% | 11.3%   | 52.9% | 85.4%  | 60.7%      |
> > > > > | Zero-Shot Predict | 62.7%    | 55.8%| 86.8% | 9.9%    | 53.6% | 81.7%  | 63.6%      |
> > > > > | Zero-Shot CoT     | 68.5%    | 55.8%| 89.0% | 22.5%   | 60.1% | 85.9%  | 65.9%      |
> > > > > | BFRS              | 68.5%    | 56.3%| 90.0% | 20.0%   | 60.1% | 86.2%  | 68.5%      |
> > > > > | MIPROv2           | 56.5%    | 52.7%| 90.8% | 21.0%   | 62.0% | 86.2%  | 65.3%      |
> > > > >
> > > > > Table 3: Average accuracy (%) across 5 prompting methods and 7 benchmarks for **Qwen3-4B**.
> > > > >
> > > > > | Prompting Method   | MMLU-Pro | GPQA | GSM8K | MedCalc| Medec | HeadQA | MedBullets |
> > > > > |--------------------|----------|------|-------|---------|-------|--------|------------|
> > > > > | HELM Baseline      | 44.9%    | 34.3%| 80.2% | 4.7%    | 52.1% | 76.7%  | 41.9%      |
> > > > > | Zero-Shot Predict | 41.7%    | 35.2%| 84.1% | 11.5%   | 52.3% | 76.9%  | 43.5%      |
> > > > > | Zero-Shot CoT     | 66.2%    | 50.0%| 88.7% | 20.8%   | 53.6% | 80.9%  | 52.6%      |
> > > > > | BFRS              | 68.6%    | 47.8%| 91.9% | 22.7%   | 56.4% | 82.5%  | 51.9%      |
> > > > > | MIPROv2           | 68.6%    | 52.2%| 90.7% | 21.5%   | 56.6% | 82.5%  | 50.3%      |

---

> > > > > > ### Author Response · Authors · 2026-01-21
> > > > > > **Rebuttal by Authors (Continued)**
> > > > > >
> > > > > > **Non-critical (would strengthen)**
> > > > > >
> > > > > > **R3.C6**: *Expand task diversity to include open-ended generation and longer-context tasks, where prompt structure often has different effects than multiple-choice.*
> > > > > >
> > > > > > **Author Response**: We thank the reviewer for this suggestion and agree that understanding the impact of structured prompting on open-ended generation is an important direction. In this work, however, we intentionally focus on multiple-choice and short-answer benchmarks, where evaluation metrics are deterministic and reproducible, allowing us to isolate the effect of prompting on leaderboard outcomes.
> > > > > >
> > > > > > Evaluating open-ended generation introduces additional technical challenges. In practice, such benchmarks typically rely on learned or LLM-based evaluators (e.g., in HELM), whose judgments can be sensitive to evaluator design choices, implementation details, and randomness. Using such evaluators for prompt optimization further requires access to the same evaluation model and configuration at training time; even small discrepancies can lead to optimization-evaluation mismatch and inconsistent scoring of similar generations. These issues complicate attribution of performance changes to prompting alone and make fair, reproducible leaderboard comparisons substantially harder.
> > > > > >
> > > > > > Moreover, prior work [3] has shown that chain-of-thought prompting yields its most reliable gains on math and symbolic reasoning tasks, and is not typically used in long-form or free-response settings, where improvements are mixed or limited [3]. This suggests that open-ended generation tasks constitute a qualitatively different evaluation regime that may require different prompting interfaces.
> > > > > >
> > > > > > Our current study aims to make a precise point: under reproducible evaluation settings, leaderboard rankings can be sensitive to prompting choices. In the revised paper, we will expand the Limitations section to explicitly discuss this as a limitation and outline it as a promising direction for future work.
> > > > > >
> > > > > > [3] Sprague, Zayne Rea, et al. "To CoT or not to CoT? Chain-of-thought helps mainly on math and symbolic reasoning." The Thirteenth International Conference on Learning Representations.
> > > > > >
> > > > > > ---
> > > > > >
> > > > > > **R3.C7**: *More rigorous cost analysis: report total optimization cost (tokens and dollars) rather than citing "DSPy documentation range," and present amortization assumptions explicitly.*
> > > > > >
> > > > > > **Author Response**: We thank the reviewer for this suggestion. While we agree that reporting exact optimization-time costs would be ideal, we did not retain token-level logs for all optimization runs in the original sweep. That said, we verified via repeat runs that the optimization cost was <$10 even in our most expensive configuration, and negligible relative to the dominant cost of full benchmark evaluation across multiple models and benchmarks. For this reason, we focus on inference-time token overhead, which is the recurring cost relevant to leaderboard reporting and deployment. We will clarify this tradeoff and the amortization assumption more explicitly in the revised manuscript.
> > > > > >
> > > > > > ---
> > > > > >
> > > > > > **R3.C8**: *Rank stability analysis: compute rank distributions under bootstrap resampling and (if possible) repeated runs, not just point ranks.*
> > > > > >
> > > > > > **Author Response**: We thank the reviewer for this suggestion. In response, we conducted a rank stability analysis using bootstrap resampling at the instance level. Specifically, for each benchmark, we resample evaluation instances with replacement, recompute model accuracies, convert them to per-benchmark ranks, and aggregate mean ranks across benchmarks. Repeating this procedure yields a bootstrap distribution over mean ranks for both the baseline and structured-prompt settings.
> > > > > >
> > > > > > The analysis shows that while the 95% bootstrap intervals overlap in cases, the direction of the rank shift is stable across resamples, and we now report bootstrap mean ranks with 95% CIs to make this uncertainty explicit. For example, under bootstrap resampling, Claude 3.7 Sonnet’s expected mean rank improves from 2.27 [1.86, 2.71] under the baseline prompt to 2.05 [1.71, 2.43] under structured prompting, indicating a consistent improvement. We will update the paper to report bootstrap mean ranks with 95% confidence intervals instead of relying solely on point rank estimates.

---

> > > > > > > ### Author Response · Authors · 2026-01-21
> > > > > > > **Rebuttal by Authors (Continued)**
> > > > > > >
> > > > > > > **R3.C9**: *Ablate the structured format vs CoT content: show whether the improvement comes from "REASONING/OUTPUT fields" formatting, from CoT instruction alone, or from other DSPy template choices.*
> > > > > > >
> > > > > > > **Author Response**: We thank the reviewer for this suggestion. In fact, our experimental design already includes a targeted ablation that isolates the contribution of CoT content from DSPy’s structured template. Moving from HELM’s baseline prompt to DSPy Zero-Shot Predict, where the primary change is the standardized DSPy prompt wrapper without CoT, yields minimal improvement when averaged across benchmarks and models (64.6% → 64.9%). In contrast, introducing a simple CoT instruction (DSPy Zero-Shot CoT) produces the dominant performance gain (64.9% → 68.5%). This demonstrates that the primary source of improvement arises from DSPy’s CoT interface rather than DSPy’s structured template.
> > > > > > > While finer-grained ablations of individual prompt fields (e.g., REASONING/OUTPUT headers) vs CoT instruction could be informative, we explicitly identify this as a limitation and a direction for future work.
> > > > > > >
> > > > > > > ---
> > > > > > >
> > > > > > > **Broader Impact Concerns**
> > > > > > >
> > > > > > > **R3.C10**: *Leaderboard gaming: The paper encourages optimizing prompts for benchmarks and reporting "ceilings." This can further incentivize benchmark overfitting and reduce the value of leaderboards as measures of real-world performance unless strict protocols are adopted (e.g., restricted prompt search budgets, standardized reporting of prompt families). This should be explicitly discussed as a potential negative externality.*
> > > > > > >
> > > > > > > **Author Response**: We thank the reviewer for raising this important concern and agree that unrestricted prompt optimization could incentivize leaderboard gaming and reduce the interpretability of benchmark results. We will add an explicit discussion of this risk in the Limitations / Broader Impact section of the revised manuscript. In particular, we will discuss the need for clearer evaluation protocols, such as standardized prompt families, reporting of prompt budgets, or multiple-prompt summaries, to mitigate these negative externalities.
> > > > > > >
> > > > > > > ---
> > > > > > >
> > > > > > > **R3.C11**: *Medical prompting risks: Some prompts use high-stakes framing ("busy emergency room," "patient’s life may depend on your accuracy"). If reused outside benchmarking, this can encourage overconfident outputs. The broader impact section should caution against deploying such prompting patterns in clinical settings without safeguards.*
> > > > > > >
> > > > > > > **Author Response**: We agree that high-stakes framing in prompts (e.g., emergency or life-critical scenarios) can encourage overconfident outputs if deployed outside controlled evaluation settings. Our use of such framing is strictly limited to benchmarking consistency and does not constitute a recommendation for real-world clinical deployment. In the revised manuscript, we will expand the Broader Impact / Limitations section to explicitly caution against reusing these prompting patterns in clinical or decision-making settings without appropriate safeguards.

---

### Review · Reviewer_2j6k · 2025-12-22

**Summary Of Contributions:**

The paper applies prompt optimization techniques to test and compare different LLM families on HELM benchmark tasks.
The paper finds that with some prompt optimization techniques, you are able to
(1) Achieve higher performance on HELM benchmark,
(2) Achieve lower variance in performance,
(3) Flip some of the existing baseline rankings,
And the additional benefits of prompt optimization saturate with reasoning/CoT prompting.

Strengths:
- Extensive experiments that span across different tasks and model families, with various prompt optimization techniques.

Weaknesses:
- Several unnecessary content and lack of substance
- The theoretical analysis is underdeveloped and not organized
- Lack of rigor in experiment result analysis

**Audience:**

No

**Audience Explanation:**

The main findings re-verified in this paper are already well-known facts within the LLM community: (1) LLMs generally do better with CoT vs non-CoT, and (2) LLMs are sensitive to prompt optimization. While the paper contains several experimental results that partially support this claim, the element of knowledge and interest that can be shared with the LLM community looks quite limited.

**Broader Impact Concerns:**

No impact or ethical concerns.

**Claims And Evidence:**

No

**Claims Explanation:**

- The core idea of the paper is the robustness of evaluation achieved through structured prompting. For instance, Table 5 caption claims that “structured prompting consistently improve robustness of benchmarks”. What is the "robustness" achieved here? The stddev values look pretty much the same for baseline vs structured prompts; one baseline performance is better vs structured prompts (Claude 3.7, Medec); many performance gaps lie well within the confidence intervals. How does this result show robustness of the evaluation?
- Reduced across-benchmark variance: is this a statistically significant drop?
- CoT reduces sensitivity to prompt design: this is a statement made from observing three prompt engineering approaches (Zero-Shot CoT, BFRS, MIPROv2). To make such a statement, more diverse prompt templates need to be tested. Even some results do seem to have some sensitivity even with CoT (e.g. GPT4o performance for MMLU-Pro in Table 4). What happens if the prompts are not optimized but are simply generated to sound reasonable? Would this still have a similar performance around Zero-Shot CoT, BFRS, or MIPROv2?
- Related to this, in Section 3.2, why doesn’t $\tau$ (reasoning traces) vary much with different prompt designs? I think this is the core logical step that needs to be proven but is not shown.

**Requested Changes:**

(Critical) Please make the evaluation and analysis of the experimental results more rigorous
- What is “robustness” that is achieved through prompt optimization? Clearly define what this is and back it up with the results
- Proper statistical testing for performance increases/decreases, variance increase/decreases

(Critical) The theory in Section 3.2 needs improvement in terms of clarity, structure, and completeness. Otherwise, it adds no value to the paper’s overall thesis and is better to just remove.
- Please state the main statement that is being proven in this section. State that in a theorem, provide a proof. If additional things need to be proven to prove that theorem, define lemmas and provide their proofs accordingly. Tie them all together in an organized, logical way.
- It states that "once COT is enabled, the effective KL divergence between path distributions under different structured prompts is small enough that equation 11 holds for most items”. Further justification behind this is needed: Why doesn’t $\tau$ vary much with different prompt designs? Why is the KL divergence small? Is this shown in the experimental results (e.g. particular values of KL divergence)? Either theoretical or empirical evidence for this statement must be shown.

(Critical) The paper also contains several unnecessary details in the main body. Things that should be moved to the appendix or removed:
- BFRS, MIPROv2 Algorithms: this takes up an entire page with no substance added to the content. These are pre-existing algorithms which can be referred to as a single line in the body with corresponding citations. The details of the algorithm plays no important role in the development of the paper, so I recommend leaving this out or moving it to the appendix.
- Equation 1 and the "formal" description of DSPy: also unnecessary like the above.
- Section 3.3: the cost analysis feels a little tangential to the main thesis of the paper. Nevertheless, it still is a necessary part to check as CoT/prompt optimization can take up more cost. This can move to the appendix and the body can refer to it for the interested readers. If this is to be kept in the main body, the cost aspect needs to be mentioned more and emphasized in the abstract and introduction – for instance, what is the method that performed the best in terms of cost performance trade-off? What are some consideration to make for best trade-offs? etc.
- Appendix: the appendix is not referred to at all in the main paper. Also, this is a description of the pseudocode of the two algorithms. It is redundant content that adds no substance to the paper and should be removed.

---

> ### Author Response · Authors · 2026-01-21
> **Rebuttal by Authors**
>
> **R2.C1**: *The main findings re-verified in this paper are already well-known facts within the LLM community: (1) LLMs generally do better with CoT vs non-CoT, and (2) LLMs are sensitive to prompt optimization. While the paper contains several experimental results that partially support this claim, the element of knowledge and interest that can be shared with the LLM community looks quite limited.*
>
> **Author Response**: We thank the reviewer for the detailed feedback and agree that chain-of-thought and prompt sensitivity are broadly recognized phenomena. While prompt sensitivity is acknowledged in principle, we believe there is a gap between general awareness and systematic quantification within widely used benchmarks. We would like to clarify our contributions and how our findings extend beyond these established observations.
>
> 1. Our goal is not to show that CoT improves performance; this is well known. Rather, our contribution is to systematically study structured prompting methods within a widely adopted, reproducible leaderboard framework and quantify their impact on reported performance and rankings. While prior prompting work typically evaluates methods in bespoke experimental setups, HELM provides a standardized and reproducible evaluation that isolates the effect of how models are invoked. By integrating structured prompting into this established framework, we show how prompting alone can materially change leaderboard outcomes. A key and non-obvious finding is that once CoT is introduced, more sophisticated prompt optimizers (BFRS, MIPROv2) yield virtually no additional gain (e.g., 68.5% → 68.6% on average), indicating a saturation effect. This has practical implications: performance gains can be achieved with simple Zero-Shot CoT, without costly prompt optimization.
> 2. In addition to the three instances of ranking flips reported in our original submission, we observe similar behavior when examining open-source models across different scales in new experiments. On MedCalc-Bench, under the HELM baseline prompt, Llama-3.3-70B substantially outperforms Qwen3-4B. However, when evaluated at ceiling (best among evaluated prompting strategies), Qwen3-4B not only closes the gap but slightly outperforms Llama-3.3-70B, despite being roughly 18x smaller. This further reinforces our central claim: fixed-prompt leaderboards can underrepresent performance, and reported rankings may reflect prompt choice rather than capability.
> 3. Beyond empirical findings, we contribute a reproducible DSPy+HELM integration and open-source prompt optimization pipeline, enabling the community to replicate, audit, and extend our analysis to additional benchmarks and models.
>
> ---
>
> **R2.C2**: *CoT reduces sensitivity to prompt design: this is a statement made from observing three prompt engineering approaches (Zero-Shot CoT, BFRS, MIPROv2). To make such a statement, more diverse prompt templates need to be tested. Even some results do seem to have some sensitivity even with CoT (e.g. GPT4o performance for MMLU-Pro in Table 4). What happens if the prompts are not optimized but are simply generated to sound reasonable? Would this still have a similar performance around Zero-Shot CoT, BFRS, or MIPROv2?*
>
> **Author Response**: Regarding the reviewer’s question about “reasonable but unoptimized” prompts: Zero-Shot CoT already represents such a setting. It uses a minimal, generic chain-of-thought instruction without task-specific optimization or demonstrations. Across models and benchmarks, this simple CoT prompt approaches performance comparable to more sophisticated optimizers, indicating that strong performance does not depend on carefully engineered or optimized templates.
>
> We would further like to clarify that our goal is not to eliminate prompt sensitivity; some variability remains even among CoT-based methods (e.g., GPT-4o on MMLU-Pro). Rather, we claim and hope to achieve a lower sensitivity than just the baseline prompting method. Once CoT is enabled, performance gains largely saturate across diverse structured prompts, suggesting that evaluation outcomes become less dependent on prompt optimization choices compared to fixed, non-CoT baselines.

---

> > ### Author Response · Authors · 2026-01-21
> > **Rebuttal by Authors (Continued)**
> >
> > **R2.C3**: *(Critical) Please make the evaluation and analysis of the experimental results more rigorous. What is “robustness” that is achieved through prompt optimization? Clearly define what this is and back it up with the results. The core idea of the paper is the robustness of evaluation achieved through structured prompting. For instance, Table 5 caption claims that “structured prompting consistently improve robustness of benchmarks”. What is the "robustness" achieved here? The stddev values look pretty much the same for baseline vs structured prompts; one baseline performance is better vs structured prompts (Claude 3.7, Medec); many performance gaps lie well within the confidence intervals. How does this result show robustness of the evaluation?*
> >
> > **Author Response**: We thank the reviewer for this thoughtful critique and apologize for the ambiguity in our original use of the term robustness. We agree that our earlier wording conflated several distinct concepts, and we will revise the manuscript to clarify definitions, claims, and evidence accordingly.
> >
> > *Clarification regarding “Robustness”*
> >
> > In this work, we use evaluation robustness to mean stability of reported performance estimates and rankings with respect to reasonable prompt choices, rather than variance reduction or uniform performance improvement. Under this definition, an evaluation is robust if its conclusions (headline performance and relative rankings) do not hinge on an arbitrary single prompt, but instead approximate a model’s attainable capability stably.
> >
> > We agree with the reviewer that the confidence intervals and across-benchmark standard deviations reported in Tables 3-5 do not demonstrate robustness. Their purpose is solely to assess whether observed performance differences are meaningful rather than due to noise. As the reviewer correctly notes, the fact that these intervals remain similar across prompting methods indicates that the observed gains are not driven by increased variance; this supports statistical significance, not robustness. These findings are further supported by paired statistical significance tests, described in detail later, which show that the aggregate performance gaps induced by prompt choice are statistically significant at the instance level and unlikely to arise by chance. We regret the confusion and will revise table captions and surrounding text to make this distinction explicit.
> >
> > Our claim rests on three observations:
> > 1. *Large performance shifts when moving from non-CoT to CoT prompts indicate the relative instability of fixed, non-CoT evaluation*: A simple, generic CoT instruction, without task-specific optimization, produces substantial performance changes, showing that fixed non-CoT prompts can underrepresent model capability and yield rankings that are sensitive to incidental prompt choice. In contrast, once a CoT interface is introduced, performance stabilizes across different structured prompting variants, and comparable prompt changes no longer induce large shifts.
> > 2. *Convergence within CoT-based prompting indicates increased stability*: Once CoT is introduced, performance estimates stabilize: Zero-Shot CoT, BFRS, and MIPROv2, despite substantial differences in structure, yield similar results (e.g., 68.5% → 68.6% on average). The absence of large swings within CoT-based methods contrasts sharply with the large jump from non-CoT to CoT, suggesting reduced sensitivity relative to baseline, under the prompt families we tested.
> > 3. *Model-specific evidence (e.g., o3-mini) reinforces this pattern*: For the reasoning-optimized model o3-mini, performance deltas across all prompting strategies (CoT or non-CoT) are consistently small, indicating less sensitivity to prompt formulation. This further supports the view that CoT-based evaluation can yield more stable estimates of model capability.

---

> > > ### Author Response · Authors · 2026-01-21
> > > **Rebuttal by Authors (Continued)**
> > >
> > > **R2.C3 (CONTINUED)**
> > >
> > > **Author Response**:
> > >
> > > *New Experiment 1: Controlled Prompt Variation Analysis*
> > >
> > > While we believe broad prompt-perturbation studies are an important direction for future work, we conducted a controlled prompt variation analysis to isolate the effect of enabling CoT under otherwise matched prompt content. We compared:
> > > 1. Non-CoT Variants: (i) Zero-Shot HELM Baseline, (ii) In-Context Demonstrations without CoT, (iii) Optimized Instructions with Demonstrations but without CoT
> > > 2. CoT Structured Variants: (i) Zero-Shot CoT, (ii) BFRS, (iii) MIPROv2
> > >
> > > Importantly, the Non-CoT variants were constructed to match the CoT-based prompts in instructions and demonstrations, differing only in the absence of the DSPy CoT interface. This controlled experiment isolates the impact of CoT itself, allowing us to assess prompt sensitivity under comparable prompt complexity.
> > >
> > > For MedCalc-Bench, we conducted a focused subset analysis using two representative models, a widely used frontier model (GPT-4o) and a smaller open-source model (Qwen3-4B), to examine sensitivity to structured prompt variation across model scales. For Qwen3-4B, Non-CoT prompt variants yield a mean±stdev accuracy of 8.7% ± 3.5%, whereas CoT-based structured prompts yield 21.7% ± 1.0%, indicating both substantially higher performance and markedly lower variance under CoT. Similarly, for GPT-4o, Non-CoT variants achieve 19.9% ± 1.0%, while CoT-based prompts reach 26.8% ± 0.2%, again exhibiting reduced sensitivity relative to baseline, under the prompt families we tested. While limited in scope, we hope this analysis provides more concrete support for our claim. Accordingly, we will revise the introduction, discussion, and table captions to reflect our more precise claim and explicitly frame broader prompt sensitivity analysis as an important direction for future work.
> > >
> > > *New Experiment 2: Open-Source Models for Experimental Rigor*
> > >
> > > Furthermore, we have expanded the experimental section to include two open-source models: Llama-3.3-70B-Instruct and Qwen3-4B-Instruct-2507, which represent complementary points in the open-source landscape, one large, frequently reported model on public benchmarks, and one smaller model known for strong performance relative to its parameter count. The new experiments  (Table 1 and 2) are consistent with our main findings and support the generality of our conclusions.
> > >
> > > 1. Structured prompting methods consistently outperform HELM’s fixed baseline, mirroring the trends observed for frontier models.
> > > 2. Across Llama-3.3-70B and Qwen3-4B, introducing chain-of-thought (Zero-Shot CoT) yields the majority of performance gains, while more sophisticated optimizers (BFRS, MIPROv2) provide limited additional benefit.
> > > 3. On MedCalc-Bench, when evaluated at ceiling (best among evaluated prompting strategies), Qwen3-4B not only closes the gap but slightly outperforms Llama-3.3-70B, despite being roughly 18x smaller. This further reinforces our central claim: fixed-prompt leaderboards can underrepresent performance, and reported rankings may reflect prompt choice rather than capability.
> > >
> > > Table 1: Average accuracy (%) across 5 prompting methods and 7 benchmarks for **Llama3.3-70B**.
> > >
> > > | Prompting Method   | MMLU-Pro | GPQA | GSM8K | MedCalc | Medec | HeadQA | MedBullets |
> > > |--------------------|----------|------|-------|---------|-------|--------|------------|
> > > | HELM Baseline      | 64.7%    | 57.0%| 85.1% | 11.3%   | 52.9% | 85.4%  | 60.7%      |
> > > | Zero-Shot Predict | 62.7%    | 55.8%| 86.8% | 9.9%    | 53.6% | 81.7%  | 63.6%      |
> > > | Zero-Shot CoT     | 68.5%    | 55.8%| 89.0% | 22.5%   | 60.1% | 85.9%  | 65.9%      |
> > > | BFRS              | 68.5%    | 56.3%| 90.0% | 20.0%   | 60.1% | 86.2%  | 68.5%      |
> > > | MIPROv2           | 56.5%    | 52.7%| 90.8% | 21.0%   | 62.0% | 86.2%  | 65.3%      |
> > >
> > > Table 2: Average accuracy (%) across 5 prompting methods and 7 benchmarks for **Qwen3-4B**.
> > >
> > > | Prompting Method   | MMLU-Pro | GPQA | GSM8K | MedCalc| Medec | HeadQA | MedBullets |
> > > |--------------------|----------|------|-------|---------|-------|--------|------------|
> > > | HELM Baseline      | 44.9%    | 34.3%| 80.2% | 4.7%    | 52.1% | 76.7%  | 41.9%      |
> > > | Zero-Shot Predict | 41.7%    | 35.2%| 84.1% | 11.5%   | 52.3% | 76.9%  | 43.5%      |
> > > | Zero-Shot CoT     | 66.2%    | 50.0%| 88.7% | 20.8%   | 53.6% | 80.9%  | 52.6%      |
> > > | BFRS              | 68.6%    | 47.8%| 91.9% | 22.7%   | 56.4% | 82.5%  | 51.9%      |
> > > | MIPROv2           | 68.6%    | 52.2%| 90.7% | 21.5%   | 56.6% | 82.5%  | 50.3%      |

---

> > > > ### Author Response · Authors · 2026-01-21
> > > > **Rebuttal by Authors (Continued)**
> > > >
> > > > **R2.C4**: *Proper statistical testing for performance increases/decreases, variance increases/decreases.*
> > > >
> > > > **Author Response**: We thank the reviewer for requesting a more rigorous statistical analysis. In response, we have added paired statistical significance testing for all reported performance differences using an exact two-sided McNemar test, for paired binary outcomes (correct/incorrect) evaluated on the same instances. To account for multiple comparisons, we additionally apply a Benjamini-Hochberg false discovery rate (FDR) correction. This enables us to test whether differences between each prompting method and the HELM baseline reflect systematic per-instance improvements rather than noise.
> > > >
> > > > First, we introduce a pooled significance analysis, aggregating paired comparisons across all benchmark instances for each model. Under this pooled analysis, structured methods (Zero-Shot CoT, BFRS, and MIPROv2) yield statistically significant improvements over the baseline for all four models. This pooled result is important for our central claim: it shows that the aggregate performance gaps induced by prompt choice are statistically significant at the instance level and therefore unlikely to arise by chance.
> > > >
> > > > At the benchmark level, structured prompting methods, particularly Zero-Shot CoT, BFRS, and MIPROv2, are significantly different (p<0.05) over the baseline for the majority of benchmark-model pairs. Conversely, on benchmarks where performance differences are small (notably Medec and HeadQA), McNemar’s test typically does not find significance for structured methods, which is expected given minimal performance differences. We also observe that o3-mini exhibits fewer significant changes across prompting methods, consistent with its smaller sensitivity to prompt choice in our experiments.
> > > >
> > > > We will update Tables 3, 4, and 5 in the revised manuscript to include significance markers and add a description of the testing protocol in the experimental section.
> > > >
> > > > ---
> > > >
> > > > **R2.C5**: *(Critical) The theory in Section 3.2 needs improvement in terms of clarity, structure, and completeness. Otherwise, it adds no value to the paper’s overall thesis and is better to just remove. Please state the main statement that is being proven in this section. State that in a theorem, provide a proof. If additional things need to be proven to prove that theorem, define lemmas and provide their proofs accordingly. Tie them all together in an organized, logical way.*
> > > >
> > > > **Author Response**: We will explicitly state the main theoretical claim as a formal decision-stability theorem in the revised paper, which establishes when predictions are invariant to controlled prompt variations under CoT prompting.
> > > >
> > > > ---
> > > >
> > > > **R2.C6**: *It states that "once COT is enabled, the effective KL divergence between path distributions under different structured prompts is small enough that equation 11 holds for most items”. Further justification behind this is needed: Why doesn’t  vary much with different prompt designs? Why is the KL divergence small? Is this shown in the experimental results (e.g. particular values of KL divergence)? Either theoretical or empirical evidence for this statement must be shown.*
> > > >
> > > > **Author Response**: We will revise Section 3.2 to clarify the scope and interpretation of the theoretical result. The decision-stability theorem now explicitly provides a sufficient condition for prediction invariance under prompt variation, stated in terms of divergence between reasoning-path distributions. We do not claim to empirically estimate or bound this divergence. Instead, we relate the theorem to an observable empirical pattern: once CoT is enabled, performance varies only slightly across the CoT prompt families we evaluate (Zero-Shot CoT, BFRS, MIPROv2; e.g., mean accuracy 68.5% → 68.6%). We will explicitly state that these lower performance differences are consistent with the theorem’s sufficient condition holding for these prompt families under deterministic decoding, rather than evidence that the divergence itself is small.

---

> > > > > ### Author Response · Authors · 2026-01-21
> > > > > **Rebuttal by Authors (Continued)**
> > > > >
> > > > > **R2.C7**: *Related to this, in Section 3.2, why doesn’t  (reasoning traces) vary much with different prompt designs? I think this is the core logical step that needs to be proven but is not shown.*
> > > > >
> > > > > **Author Response**: We agree that explaining why reasoning traces remain similar across different CoT prompts is central to the argument, and we will make this explicit. Under a shared CoT interface, the model is constrained to generate step-by-step reasoning before producing an answer, which significantly reduces the space of admissible reasoning trajectories. Differences in prompt wording or demonstrations primarily affect superficial phrasing or ordering of steps rather than the underlying logical structure of the reasoning path. As a result, the high-probability reasoning traces that lead to correct answers overlap strongly across structured prompts, yielding similar reasoning-path distributions. This mechanism will be formalized in the appendix and directly tied to the decision-stability result in the main paper.
> > > > >
> > > > > ---
> > > > >
> > > > > **R2.C8**: *(Critical) The paper also contains several unnecessary details in the main body. Things that should be moved to the appendix or removed: BFRS, MIPROv2 Algorithms: this takes up an entire page with no substance added to the content. These are pre-existing algorithms which can be referred to as a single line in the body with corresponding citations. The details of the algorithm plays no important role in the development of the paper, so I recommend leaving this out or moving it to the appendix. Equation 1 and the "formal" description of DSPy: also unnecessary like the above.*
> > > > >
> > > > > **Author Response**: We thank the reviewer for this feedback and agree that these details are not essential to the main narrative. In the revised manuscript, we will move the descriptions of the BFRS and MIPROv2 algorithms, as well as Equation 1 and the formal description of DSPy, to the appendix, retaining only brief references and citations in the main body.
> > > > >
> > > > > ---
> > > > >
> > > > > **R2.C9**: *Section 3.3: the cost analysis feels a little tangential to the main thesis of the paper. Nevertheless, it still is a necessary part to check as CoT/prompt optimization can take up more cost. This can move to the appendix and the body can refer to it for the interested readers. If this is to be kept in the main body, the cost aspect needs to be mentioned more and emphasized in the abstract and introduction – for instance, what is the method that performed the best in terms of cost performance trade-off? What are some consideration to make for best trade-offs? Etc.*
> > > > >
> > > > > **Author Response**: We thank the reviewer for this helpful suggestion. We believe that computational cost is an important consideration when evaluating structured prompting methods. We intentionally retain the cost analysis in the main body because it directly supports one of our central practical conclusions: while structured prompting can substantially alter leaderboard outcomes, not all prompting strategies are equally cost-effective. Our analysis shows that Zero-Shot CoT captures most of the performance gains of more expensive prompt optimizers (BFRS, MIPROv2) while adding only a small number of additional tokens. In response to the reviewer’s feedback, we will explicitly emphasize this cost–benefit perspective in both the abstract and the introduction. We have also expanded our analysis by adding new per-benchmark accuracy vs. token cost plots, which provide a more granular view of the trade-offs: https://imgur.com/a/Jb3sulg
> > > > >
> > > > > ---
> > > > >
> > > > > **R2.C10**: *Appendix: the appendix is not referred to at all in the main paper. Also, this is a description of the pseudocode of the two algorithms. It is redundant content that adds no substance to the paper and should be removed.*
> > > > >
> > > > > **Author Response**: We thank the reviewer for this suggestion. In the revised manuscript, we will completely remove the algorithm descriptions of the BFRS and MIPROv2 from the appendix, as they are pre-existing methods and not central to our contributions. The appendix will contain two sections:
> > > > >
> > > > > 1. The full theoretical proof from section 3.2 “Why Chain-of-Thought Reduces Sensitivity to Prompt Design”.
> > > > > 2. BFRS/MIPROv2 Algorithms moved from the main text to the appendix.
> > > > >
> > > > > Furthermore, we will explicitly reference the appendix from the main paper, where it contains material relevant to the above two sections.

---

### Review · Reviewer_rM7u · 2025-12-29

**Summary Of Contributions:**

1. A reproducible “DSPy+HELM” evaluation framework that integrates structured prompting methods into the established HELM benchmark, enabling more accurate estimation of language model performance ceilings.

2. A systematic evaluation of four structured prompting methods (Zero-Shot CoT, BFRS, MIPROv2) across seven general and medical benchmarks using four frontier LMs.

3. Key empirical findings:

(1) HELM with fixed prompts systematically underestimates LM performance (by ~4% on average).

(2) Structured prompting reduces cross-benchmark variance (standard deviation decreases by ~2%).

(3) Model rankings change on several benchmarks (e.g., MMLU-Pro, GSM8K, MedCalc-Bench).

(4) Introducing chain-of-thought (CoT) significantly reduces sensitivity to prompt design.

**Audience:**

Yes

**Audience Explanation:**

1. High topical relevance:

LM evaluation and benchmarking are central to current AI research. TMLR readers are keenly interested in robust evaluation methodologies, prompt engineering, and model comparison.

2. Practical implications:

The findings reveal systemic underestimation in fixed-prompt benchmarks, which affects model selection and deployment decisions in real-world applications.

3. Methodological contribution:

The DSPy+HELM framework provides a scalable, automated approach to prompt optimization, advancing the field beyond manual prompt tuning.

4. Cross-domain applicability:

Experiments cover both general and medical tasks, making the work relevant to researchers in healthcare AI, education, legal NLP, and other specialized domains.

**Broader Impact Concerns:**

1. Fairer and more comparable evaluations:
Reduces bias introduced by arbitrary prompt choices, leading to more equitable model comparisons.

2. Advances in automated prompt engineering:
Promotes systematic, reproducible prompt optimization, moving the field away from ad-hoc manual tuning.

3. Supports responsible AI deployment:
More accurate performance estimates help practitioners make safer and more reliable model choices in high-stakes domains like healthcare.

**Claims And Evidence:**

Yes

**Claims Explanation:**

1. Rigorous experimental design:

(1) Four widely-used frontier LMs, seven diverse benchmarks spanning general and medical domains.

(2) Five prompting methods compared, including HELM baseline and three structured variants.

(3) Clear setup: deterministic sampling (temperature=0), consistent with HELM’s evaluation protocol.

2. Comprehensive results:

(1) Tables 3–5 provide detailed performance metrics (means, standard deviations, confidence intervals).

(2) Results consistently show performance gains, reduced variance, and rank changes under structured prompting.

3. Theoretical grounding:

(1) Pages 9–10 offer a formal model explaining the CoT-induced robustness, using Markov assumptions and divergence bounds.

4. Reproducibility:

(1) Code and pipeline are open-sourced, facilitating verification and extension.

**Requested Changes:**

1. Expand Generalizability Discussion
The current results focus on frontier LMs; please discuss the potential applicability and limitations for open-source and smaller models, possibly with preliminary experiments or references.

2. Broaden Task Coverage
The evaluation is limited to multiple-choice and short reasoning tasks. Consider including or discussing open-ended generation tasks to demonstrate the broader impact of structured prompting.

3. Clarify Method Selection
Only a subset of DSPy-based methods is evaluated. Please clarify the criteria for method selection and discuss how other DSPy modules or prompting strategies might perform.

4. Add More Quantitative Details
Provide additional quantitative results, such as detailed accuracy and token cost breakdowns per method and benchmark, ideally in tabular or graphical format.

5. Improve Theoretical Explanation
The Markov chain and KL divergence bound are briefly mentioned. Please elaborate on these theoretical insights with more formal definitions, proofs, or illustrative examples for clarity.

---

> ### Author Response · Authors · 2026-01-21
> **Rebuttal by Authors**
>
> We thank the reviewer for the positive and thorough assessment of our contributions, experimental rigor, and the relevance of our findings to improving language model evaluation.
>
> ---
>
> **R1.C1**: *Expand Generalizability Discussion. The current results focus on frontier LMs; please discuss the potential applicability and limitations for open-source and smaller models, possibly with preliminary experiments or references.*
>
> **Author Response**: We thank the reviewer for highlighting the importance of generalizability beyond frontier models. To clarify, our submission focuses on frontier LMs intentionally: the goal was to demonstrate that even state-of-the-art models can exhibit sensitivity to prompt design, implying that fixed-prompt leaderboards can underrepresent capability via altered performance estimates and rankings. That said, we agree that explicitly validating this behavior across open-source models would strengthen the analysis.
> In response, we have expanded the experimental section to include two open-source models: Llama-3.3-70B-Instruct and Qwen3-4B-Instruct-2507, which represent complementary points in the open-source landscape, one large, frequently reported model on public benchmarks, and one smaller model known for strong performance relative to its parameter count. The new experiments  (Table 1 and 2) are consistent with our main findings and support the generality of our conclusions.
>
> 1. Structured prompting methods consistently outperform HELM’s fixed baseline, mirroring the trends observed for frontier models.
> 2. Across Llama-3.3-70B and Qwen3-4B, introducing chain-of-thought (Zero-Shot CoT) yields the majority of performance gains, while more sophisticated optimizers (BFRS, MIPROv2) provide limited additional benefit.
> 3. On MedCalc-Bench, when evaluated at ceiling (best among evaluated prompting strategies), Qwen3-4B not only closes the gap but slightly outperforms Llama-3.3-70B, despite being roughly 18x smaller. This further reinforces our central claim: fixed-prompt leaderboards can underrepresent performance, and reported rankings may reflect prompt choice rather than capability.
>
> Table 1: Average accuracy (%) across 5 prompting methods and 7 benchmarks for **Llama3.3-70B**.
>
> | Prompting Method   | MMLU-Pro | GPQA | GSM8K | MedCalc | Medec | HeadQA | MedBullets |
> |--------------------|----------|------|-------|---------|-------|--------|------------|
> | HELM Baseline      | 64.7%    | 57.0%| 85.1% | 11.3%   | 52.9% | 85.4%  | 60.7%      |
> | Zero-Shot Predict | 62.7%    | 55.8%| 86.8% | 9.9%    | 53.6% | 81.7%  | 63.6%      |
> | Zero-Shot CoT     | 68.5%    | 55.8%| 89.0% | 22.5%   | 60.1% | 85.9%  | 65.9%      |
> | BFRS              | 68.5%    | 56.3%| 90.0% | 20.0%   | 60.1% | 86.2%  | 68.5%      |
> | MIPROv2           | 56.5%    | 52.7%| 90.8% | 21.0%   | 62.0% | 86.2%  | 65.3%      |
>
> Table 2: Average accuracy (%) across 5 prompting methods and 7 benchmarks for **Qwen3-4B**.
>
> | Prompting Method   | MMLU-Pro | GPQA | GSM8K | MedCalc| Medec | HeadQA | MedBullets |
> |--------------------|----------|------|-------|---------|-------|--------|------------|
> | HELM Baseline      | 44.9%    | 34.3%| 80.2% | 4.7%    | 52.1% | 76.7%  | 41.9%      |
> | Zero-Shot Predict | 41.7%    | 35.2%| 84.1% | 11.5%   | 52.3% | 76.9%  | 43.5%      |
> | Zero-Shot CoT     | 66.2%    | 50.0%| 88.7% | 20.8%   | 53.6% | 80.9%  | 52.6%      |
> | BFRS              | 68.6%    | 47.8%| 91.9% | 22.7%   | 56.4% | 82.5%  | 51.9%      |
> | MIPROv2           | 68.6%    | 52.2%| 90.7% | 21.5%   | 56.6% | 82.5%  | 50.3%      |

---

> > ### Author Response · Authors · 2026-01-21
> > **Rebuttal by Authors (Continued)**
> >
> > **R1.C2**: *Broaden Task Coverage. The evaluation is limited to multiple-choice and short reasoning tasks. Consider including or discussing open-ended generation tasks to demonstrate the broader impact of structured prompting.*
> >
> > **Author Response**: We thank the reviewer for this suggestion and agree that understanding the impact of structured prompting on open-ended generation is an important direction. In this work, however, we intentionally focus on multiple-choice and short-answer benchmarks, where evaluation metrics are deterministic and reproducible, allowing us to isolate the effect of prompting on leaderboard outcomes.
> >
> > Evaluating open-ended generation introduces additional technical challenges. In practice, such benchmarks typically rely on learned or LLM-based evaluators (e.g., in HELM), whose judgments can be sensitive to evaluator design choices, implementation details, and randomness. Using such evaluators for prompt optimization further requires access to the same evaluation model and configuration at training time; even small discrepancies can lead to optimization-evaluation mismatch and inconsistent scoring of similar generations. These issues complicate attribution of performance changes to prompting alone and make fair, reproducible leaderboard comparisons substantially harder.
> >
> > Moreover, prior work [1] has shown that chain-of-thought prompting yields its most reliable gains on math and symbolic reasoning tasks, and is not typically used in long-form or free-response settings, where improvements are mixed or limited [1]. This suggests that open-ended generation tasks constitute a qualitatively different evaluation regime that may require different prompting interfaces.
> >
> > Our current study aims to make a precise point: under reproducible evaluation settings, leaderboard rankings can be sensitive to prompting choices. In the revised paper, we will expand the Limitations section to explicitly discuss this as a limitation and outline it as a promising direction for future work.
> >
> > [1] Sprague, Zayne Rea, et al. "To CoT or not to CoT? Chain-of-thought helps mainly on math and symbolic reasoning." The Thirteenth International Conference on Learning Representations.

---

> > > ### Author Response · Authors · 2026-01-21
> > > **Rebuttal by Authors (Continued)**
> > >
> > > **R1.C3**: *Clarify Method Selection Only a subset of DSPy-based methods is evaluated. Please clarify the criteria for method selection and discuss how other DSPy modules or prompting strategies might perform.*
> > >
> > > **Author Response**: We thank the reviewer for raising this point and appreciate the opportunity to clarify our experimental choices. We selected the prompting methods based on two criteria:
> > > 1. Our goal is not to benchmark or advocate for any specific evaluation framework (e.g., HELM) or optimization framework (e.g., DSPy). Rather, our central claim is framework-agnostic: leaderboard rankings are sensitive to prompting choices, and leaderboards often reflect these decisions rather than the inherent capabilities of the models. HELM and DSPy are used because they are well-established and widely adopted in the community, and because they provide a transparent and reproducible way to vary prompting strategies in a controlled manner.
> > > 2. We choose DSPy also because it is not a single optimization method but a framework that enables movement along a spectrum of commonly used prompting regimes, ranging from low-adaptation to higher-adaptation settings (e.g., Zero-Shot Predict, Zero-Shot CoT, Few-Shot BFRS, Few-Shot MIPROv2). These prompting strategies are routinely reported across literature [2], representing standard choices. Accordingly, our intent is not to exhaustively evaluate all DSPy modules or prompting strategies, but to answer a targeted question: “What happens to leaderboard rankings when a different, but standard, prompting method is used?”
> > >
> > > Further, to address the concern that only a subset of DSPy-based methods was evaluated, we also ran GEPA [2], a recent prompt optimization algorithm that has been shown to outperform reinforcement-learning-based approaches in prior work. Importantly, we use the standalone GEPA implementation, not a DSPy instantiation, to ensure that our findings are not an artifact of DSPy itself. On MMLU-Pro, GEPA achieves performance comparable to existing DSPy optimizers across models. These results (Table 3) reinforce that the precise choice of optimizer is less important than the fact that alternative prompting strategies can yield performance substantially higher than currently reported leaderboard baselines. In the revised paper, we will explicitly add both (i) a discussion clarifying the rationale for method selection and (ii) the new GEPA results (including the MMLU-Pro table).
> > >
> > > [2] Agrawal, Lakshya A., et al. "Gepa: Reflective prompt evolution can outperform reinforcement learning." arXiv preprint arXiv:2507.19457 (2025).
> > >
> > > Table 3: Average accuracy (%) across 6 prompting methods and 6 language models, including **GEPA** optimizer.
> > >
> > > | Prompting Method      | Claude 3.7 | Gemini 2.0 | GPT-4o | o3 Mini | Llama3.3 | Qwen3 |
> > > |-----------------------|------------|------------|--------|---------|----------|-------|
> > > | HELM Baseline         | 76.3%      | 66.1%      | 62.2%  | 77.1%   | 64.7%    | 44.9% |
> > > | Zero-Shot Predict    | 77.7%      | 70.3%      | 60.7%  | 78.4%   | 62.7%    | 41.7% |
> > > | Zero-Shot CoT        | 79.7%      | 75.3%      | 67.6%  | 76.2%   | 68.5%    | 66.2% |
> > > | BFRS                 | 80.1%      | 75.4%      | 71.1%  | 76.5%   | 68.5%    | 68.6% |
> > > | MIPROv2              | 80.6%      | 75.3%      | 68.7%  | 76.1%   | 56.5%    | 68.6% |
> > > | **GEPA**              | **79.0%**  | **69.5%**  | **73.2%** | **77.6%** | **63.4%** | **55.1%** |
> > >
> > > ---
> > >
> > > **R1.C4**: *Add More Quantitative Details Provide additional quantitative results, such as detailed accuracy and token cost breakdowns per method and benchmark, ideally in tabular or graphical format.*
> > >
> > > **Author Response**: We thank the reviewer for this suggestion. To provide more granular quantitative detail, we extend our analysis beyond the aggregate accuracy-cost tradeoff shown in the original figure by adding per-benchmark accuracy vs. token cost plots. Each plot reports, for a single benchmark, model accuracy as a function of additional prompt tokens relative to the HELM baseline. Across benchmarks, these plots reveal a consistent trend: Zero-Shot Predict and Zero-Shot CoT incur minimal prompt overhead, while few-shot methods such as BFRS and MIPROv2 require substantially larger token budgets. Notably, Zero-Shot CoT often achieves accuracy comparable to these more expensive methods despite being far cheaper, reinforcing the same conclusions as the aggregate analysis while directly addressing benchmark-level tradeoffs.
> > >
> > > The new per-benchmark plots will be included in the revised paper, and are also available via the following anonymous link: https://imgur.com/a/Jb3sulg

---

> > > > ### Author Response · Authors · 2026-01-21
> > > > **Rebuttal by Authors (Continued)**
> > > >
> > > > **R1.C5**: *Improve Theoretical Explanation The Markov chain and KL divergence bound are briefly mentioned. Please elaborate on these theoretical insights with more formal definitions, proofs, or illustrative examples for clarity.*
> > > >
> > > > **Author Response**: We thank the reviewer for this suggestion. In the revised paper, we substantially expand the theoretical analysis to explain why CoT reduces prompt sensitivity. Specifically, (i) we introduce a formal setup and assumptions, (ii) prove a data-processing inequality showing that output sensitivity is bounded by sensitivity in the reasoning-path distribution, and (iii) establish margin-based stability guarantees under bounded controlled prompt variations. We further (iv) derive sufficient conditions in terms of KL divergence between reasoning-path distributions and (v) connect these results to empirical evidence showing that chain-of-thought enlarges decision margins and is consistent with the sufficient conditions of the stability theorem

---

> ### Comment · Reviewer_rM7u · 2026-02-07
>
> Overall Recommendation: Accept
>
> This paper provides a timely and important contribution to the field of language model evaluation. Through systematic experiments, it compellingly demonstrates that within fixed evaluation frameworks like HELM, different prompting strategies can significantly affect estimates of model performance and ranking. Its core claims—that structured prompting (especially chain-of-thought) improves performance estimates, reduces cross-benchmark variance, and alters model rankings—are well-supported by a rigorous experimental design involving multiple frontier and open-source models.
>
> The authors have provided thorough and satisfactory responses to all reviewer comments, substantially strengthening the manuscript. The assessment is as follows:
>
> Assessment of Author Responses:
>
> Regarding Expanding Generalizability: The authors have excellently addressed this point by conducting new experiments with two open-source models (Llama-3.3-70B-Instruct and Qwen3-4B-Instruct-2507), presented in new tables. The results confirm that the core findings extend beyond frontier models, powerfully reinforcing the paper's central thesis.
>
> Status: Satisfactorily Addressed.
>
> Regarding Broadening Task Coverage: The authors provide a reasonable and convincing justification for focusing on multiple-choice and short-answer tasks, highlighting the methodological challenges of open-ended generation. Their commitment to discuss this scope in the Limitations section is acceptable.
>
> Status: Adequately Addressed.
>
> Regarding Clarifying Method Selection: This point is well addressed. The authors clarified their framework-agnostic goal and supplemented their analysis with results from a non-DSPy optimizer (GEPA), effectively demonstrating the general nature of the observed phenomenon.
>
> Status: Satisfactorily Addressed.
>
> Regarding Adding Quantitative Details: The authors have partially addressed this by generating per-benchmark accuracy vs. token cost plots. For maximum clarity, it is recommended to integrate key plots into the manuscript.
>
> Status: Addressed with Minor Revision Suggested.
>
> Regarding Improving Theoretical Explanation: The authors commit to a substantial expansion of the theoretical analysis. Acceptance should be contingent on the inclusion of this elaboration.
>
> Status: Satisfactorily Addressed (Pending Verification).
>
> Final Requested Changes for the Camera-Ready Version:
>
> Mandatory: Incorporate the elaborated theoretical analysis as promised in the response.
>
> Recommended: Integrate one or two representative accuracy/token-cost plots into the main paper or a clearly labeled appendix.
>
> Editorial: Seamlessly incorporate the new experimental results (open-source models, GEPA comparison) into the results section with appropriate commentary.
>
> The revisions have strengthened the paper, and it is now suitable for acceptance.

---

### Decision · Action_Editor_ZE2t · 2026-02-24

**Recommendation:** Reject

**Audience:**

Yes

**Audience Explanation:**

Reviewer rM7u: the work has "high topical relevance" with "practical implications" and "cross-domain applicability." Reviewer JhgW agrees: "Benchmarking methodology and evaluation validity are highly relevant to TMLR" and the DSPy+HELM integration is "still of interest to the evaluation and reliability audience." Although Reviewer 2j6k points out: "the main findings re-verified in this paper are already well-known facts within the LLM community".

**Claims And Evidence:**

No

**Claims Explanation:**

Reviewer 2j6k: "the core logic behind their theoretical result and the resulting claim remains weak and unsupported." The authors' rebuttal argument about constrained reasoning trajectories is "not a sufficiently rigorous argument." The open-source results show "MIPRO actually hurts one of the numbers significantly compared to other CoT methods, contradicting the fact that CoT performance varies only slightly across the prompt families." Reviewer JhgW: "the work still does not provide a broad, direct prompt sensitivity study across benchmarks" and the new analysis is "helpful but limited in scope." The theory-experiment gap means the theory "is not yet credible as supporting evidence."

**Resubmission Of Major Revision:**

The authors may consider submitting a major revision at a later time.